# Datasets and protocols for including anomalous freshwater from melting ice sheets in climate simulations

Gavin A. Schmidt<sup>1</sup>, Kenneth D. Mankoff<sup>2,1</sup>, Jonathan L. Bamber<sup>3,4</sup>, Clara Burgard<sup>5</sup>, Dustin Carroll<sup>6,7</sup>, David M. Chandler<sup>8</sup>, Violaine Coulon<sup>9</sup>, Benjamin J. Davison<sup>10</sup>, Matthew H. England<sup>11</sup>, Paul

R. Holland<sup>12</sup>, Nicolas C. Jourdain<sup>13</sup>, Oian Li<sup>14,15</sup>, Juliana M. Marson<sup>16</sup>, Pierre Mathiot<sup>13</sup>, Clive

R. McMahon<sup>17</sup>, Twila A. Moon<sup>18</sup>, Ruth Mottram<sup>19</sup>, Sophie Nowicki<sup>20</sup>, Anna Olivé Abelló<sup>13</sup>, Andrew

G. Pauling<sup>21</sup>, Thomas Rackow<sup>22</sup>, and Damien Ringeisen<sup>23,1,24</sup>

Correspondence: Gavin A. Schmidt (gavin.a.schmidt@nasa.gov)

**Abstract.** Anomalous freshwater fluxes from the Greenland and Antarctic ice sheets and ice shelves are impacting the surrounding oceans, and we need to be able to account for these effects in climate model simulations over the historical period and in future projections. In previous phases of the Coupled Model Intercomparison Project (CMIP), models mostly either assumed

<sup>&</sup>lt;sup>1</sup>NASA Goddard Institute for Space Studies, New York, USA

<sup>&</sup>lt;sup>2</sup>Autonomic Integra LLC, New York, USA

<sup>&</sup>lt;sup>3</sup>School of Geographical Sciences, University of Bristol, Bristol, UK;

<sup>&</sup>lt;sup>4</sup>Dept of Aerospace and Geodesy, Technical University Munich, Munich, Germany

<sup>&</sup>lt;sup>5</sup>Laboratoire d'Océanographie et du Climat Expérimentations et Approches Numériques (LOCEAN), Sorbonne Université, CNRS/IRD/MNHN, Paris, France

<sup>&</sup>lt;sup>6</sup>Moss Landing Marine Laboratories, San José State University, CA

<sup>&</sup>lt;sup>7</sup>Jet Propulsion Laboratory, California Institute of Technology, CA

<sup>&</sup>lt;sup>8</sup>NORCE Norwegian Centre, Bjerknes Centre for Climate Research, Bergen, Norway

<sup>&</sup>lt;sup>9</sup>Laboratoire de Glaciologie, Université libre de Bruxelles, Brussels, Belgium

<sup>&</sup>lt;sup>10</sup>School of Geography and Planning, University of Sheffield, Sheffield, UK

<sup>&</sup>lt;sup>11</sup>Centre for Marine Science and Innovation (CMSI), and Australian Centre for Excellence in Antarctic Science (ACEAS), University of New South Wales, Sydney 2052, NSW, Australia

<sup>&</sup>lt;sup>12</sup>British Antarctic Survey, Cambridge, UK

<sup>&</sup>lt;sup>13</sup>Univ. Grenoble Alpes, CNRS, INRAE, IRD, Grenoble INP, IGE, 38000 Grenoble, France

<sup>&</sup>lt;sup>14</sup>Department of Earth, Atmospheric, and Planetary Sciences, Massachusetts Institute of Technology, Cambridge, MA, USA 02139

<sup>&</sup>lt;sup>15</sup>Department of Earth, Ocean, and Atmospheric Science, Florida State University, Tallahassee, FL, USA 32304

<sup>&</sup>lt;sup>16</sup>Centre for Earth Observation Science, Dept. of Environment and Geography, University of Manitoba, Winnipeg, Canada

<sup>&</sup>lt;sup>17</sup>IMOS Animal Tagging, Sydney Institute of Marine Science, Mosman, 2088, Australia

<sup>&</sup>lt;sup>18</sup>National Snow and Ice Data Center, Cooperative Institute for Research in Environmental Sciences, University of Colorado, Boulder, CO, USA

<sup>&</sup>lt;sup>19</sup>Danish Meteorological Institute, Sankt Kjelds Gade 3, Copenhagen, 2100, Denmark

<sup>&</sup>lt;sup>20</sup>Department of Earth Sciences, University at Buffalo, Buffalo, NY, USA

<sup>&</sup>lt;sup>21</sup>Department of Physics, University of Otago, Dunedin, New Zealand

<sup>&</sup>lt;sup>22</sup>European Centre for Medium-Range Weather Forecasts (ECMWF), Bonn, Germany

<sup>&</sup>lt;sup>23</sup>Center for Climate Systems Research, Columbia Climate School, New York, USA

<sup>&</sup>lt;sup>24</sup>Now at Canadian Centre for Climate Modelling and Analysis (CCCma), Environment and Climate Change Canada (ECCC), Victoria, British Columbia, Canada.

that the ice sheets were in mass balance, or that discharge from the ice sheets was constant, but in neither case was the observed increasing discharge over the historical period properly represented. In this paper, we present data products of absolute and anomalous freshwater mass fluxes from both major ice sheets, and recommendations for their use in historical simulations. These fluxes can be implemented in climate simulations as a forcing for models that do not (yet) include interactive ice sheets, or used to evaluate models that do. We also make recommendations for how climatological and anomalous fluxes can be implemented in climate models that may have different approaches to interactions with the ice sheets. These forcings are available for CMIP7 simulations and should lead to more robust and coherent simulation of sea surface temperature, sea ice and regional sea level trends in the recent historical period and, as these data are extended, improve the credibility of projections.

## 1 Introduction

The loss of ice in the Earth's cryosphere has been some of the most persuasive evidence of climatically important warming over the last century and has accelerated in recent decades. The visually obvious retreat of mountain glaciers (World Glacier Monitoring Service (WGMS), 2023; Zemp et al., 2025), mass loss from the large ice sheets in Greenland and Antarctica (Otosaka et al., 2023a), and the remote sensing evidence for ice shelf changes and sea ice loss (Slater et al., 2021), are testament to ongoing and pervasive changes to the planet's climate.

Collective efforts to understand the causes of these changes and to project what impacts may come in the future have been underway since the 1980s, and since 1995 these efforts have been coordinated by the Coupled Model Intercomparison Project (CMIP) (e.g., Eyring et al., 2016), which in 2025 has entered its 7th phase (CMIP7) (Dunne et al., 2024). The climate models used in these exercises must be sufficiently computationally cheap that they can simulate tens of thousands (or even hundreds of thousands) of model years over a few real years. This limits both the resolution at which they are run, and the completeness of system physics included in the simulations. In particular, while all such models have interactive ocean and sea ice components, and land snow modules, no models in CMIP6 or previous phases included interactive ice sheets or ice shelves (referred to here as an Ice Sheet Model or ISM). The implicit assumption was that these changes in the cryosphere did not matter much for the broader climate over the recent past, except for their effect on global sea level rise which could be assessed independently (Dieng et al., 2017; Barnoud et al., 2021).

However, recent observational evidence of Southern Ocean freshening and cooling, local sea level tendencies, and (until 2015) increases in Southern Ocean sea ice concentration, has led to speculation that the impacts of changes in freshwater (FW) amounts from the ice sheets may be impacting Southern Ocean properties (Bintanja et al., 2013; Jullion et al., 2013; Rye et al., 2014; Bronselaer et al., 2018; Rye et al., 2020; Jacobs et al., 2022). This has led modelers to experiment with simulations that include the impact of freshwater changes in some models (e.g., Gomez et al., 2015; Bakker et al., 2016; Pauling et al., 2016; Merino et al., 2018; Golledge et al., 2019; Li et al., 2023a, b; Ackermann et al., 2024; Mackie et al., 2020a, b; Wagner and Eisenman, 2017; Swart et al., 2023). In the North Atlantic, the potential impact of recently increasing freshwater from Greenland and surrounding glaciers and ice caps on ocean convection and the Atlantic Meridional Overturning Circulation (AMOC) has also been recognized (Yang et al., 2016; Böning et al., 2016; Pontes and Menviel, 2024).

Past CMIP-class models and new models in development have taken a variety of approaches to the climate interaction with land ice and ice shelves. There are a number of challenges, for instance, insufficient horizontal resolution to capture mountain glaciers at high elevations, ocean model grids that ignore the ocean cavities beneath floating ice shelves, the very small scales of ice sheet dynamics, and the difficulty of initializing a stable *and* realistic pre-industrial ice-sheet state. From a partial survey of modeling groups conducted by the authors in January 2024, none of those models in CMIP6 included interactive ice sheets, or sub-ice ocean cavities or had sufficient resolution to resolve most mountain glaciers (see Appendix B).

Recently, efforts to use the historical record of changing freshwater (FW) (Slater et al., 2021) supported suggestions that the magnitude of the observed FW is potentially a first order impact on regional oceanographic conditions (Schmidt et al., 2023). This evidence strongly indicates that modeling groups need to move towards coupled earth system models and ice sheet models (ESM-ISMs) for ocean circulation, stratification and sea level purposes. However, this has been more challenging than was anticipated two decades ago (Little et al., 2007; Nowicki et al., 2016). Some groups have successfully demonstrated such capability in some model versions (e.g., Barbi et al., 2014; Muntjewerf et al., 2021; Smith et al., 2021; Siahaan et al., 2022; Goelzer et al., 2025), some of which will be used in CMIP7, but it is likely that most CMIP7 model simulations will not include this functionality.

Given this background, a virtual workshop was organized in February 2024 (Schmidt et al., 2024) to assess the potential for providing a more definitive set of observation-based freshwater volume flow rates from the ice sheets and ice shelves, and for providing guidance on how such freshwater might be used within existing climate models, given the diverse range of existing approaches (Goddard Institute for Space Studies, 2024). This paper presents the products that resulted from that workshop. We discuss the modeling needs in Section 2, while the details of the available observations and methodology to create the time-series are in Section 3. The discussion of the implementation is in Section 4. We summarize our recommendations across the experiments proposed for CMIP7 in Section 5, and add some discussion and concluding remarks in Section 6. All code and data are available through the project GitHub repository https://github.com/NASA-GISS/freshwater-forcing-workshop/ and through this Zenodo link; https://doi.org/10.5281/zenodo.14020895 (Mankoff et al., 2025a).

#### 1.1 Definitions and Assumptions

60

65

There is often a lack of clarity in how the various processes by which freshwater mass from land ice reach the ocean, and so we provide some working definitions here so that they can be used consistently in this paper and the accompanying datasets.

- Calving the process that generates icebergs, regardless of their size.
- **Discharge** represents ice flux through "gates" upstream of, or at, an ice sheet grounding line. **Discharge** leads to calving, frontal melt, and sub-shelf melt of floating ice shelves. This term is only pertinent to marine-terminating glaciers.
- Freshwater flux is used here to represent the water mass flow reaching the ocean in the form of icebergs, sub-shelf
  melt, frontal melt, or runoff in terms of mass units per time.

**Figure 1.** Schematic of the key processes contributing to freshwater fluxes from the ice sheets and ice shelves to the ocean, including some of the asymmetries between Greenland and Antarctica.

- Grounded ice total amount of ice upstream of the final grounding line. If this ice discharges across the final grounding line it displaces ocean water and contributes to (barystatic) sea level rise and, if it melts, ocean freshening.
- Ice front marks the dynamic edge of the ice shelf or marine terminating glacier that is physically connected to the ice sheet. It can extend tens of meters above the ocean surface to hundreds of meters below.
  - Ice shelf Meteoric glacier ice that is floating on the ocean but still connected to the ice sheet. Some ice shelves include
    a layer of marine ice at their base, formed when melt water from deeper ice cools below the pressure-dependent freezing
    point.
- Iceberg melt freshwater addition from icebergs and its vertical and horizontal distribution in the ocean. This is made
   up of melt that occurs in the ocean, as well as a (very small) amount of surface runoff from the protruding mass.
  - Runoff melted ice from the surface of an ice sheet, glacier, or ice shelf, that does not refreeze or is otherwise retained, plus melted snow and rain on an ice sheet or surrounding land. Runoff is a mass loss process where water is added to the ocean or water bodies on land. It can be routed at the surface, through the glacier, or at the bed of the ice sheet and occurs for both marine and land-terminating ice margins. In this paper, we refer mainly to surface runoff, and assume that that basal runoff from underneath grounded ice is negligible. Note that runoff is assumed to be independently calculated by the class of models discussed here.

- **Sub-shelf melt** - the net amount of melting at the bottom of an ice shelf. (There is also some refreezing of ocean water possible at the base of ice shelves).

Submarine melt - the net total amount of marine melt, which consists of melt at the ice front, and at the base of the ice shelf.

85

100

105

- Surface Mass Balance (SMB) - Locally this is the net effect of surface mass fluxes (SMB = Pred - Evap - Runoff). When we discuss the net ice sheet SMB, it is the integral over the total surface area of the ice sheet.

In modeling this freshwater, we need to be clear which domain it is applied on, which can vary in different applications or regionally. Many ocean models do not include the cavities underneath the large ice shelves (none in the CMIP6 models surveyed in the Appendix) and so have a boundary at the ice front. Similarly, many fjords around Greenland and elsewhere are not resolved, and so the domain boundary is assumed to be at the fjord mouth, downstream of the actual observed calving front, which can be more than 100 km away. Models may also have hybrid situations, for instance, including the Ross Sea and Weddell Sea sub-ice shelf cavities, but not elsewhere (Hutchinson et al., 2023) and with severely reduced representation of sub-grid scale features such as basal channels incised on the underside of ice shelves. In the implementation section, we will discuss the implications of these variations in ocean model practice.

The freshwater can additionally be broken down into a climatological (constant, perhaps seasonally-varying) amount, and an anomalous, time-varying, component. Depending on the application and the model structure, users may want to use the total mass flow or just the anomalies. Different observational products are more suited to calculating anomalies (such as GRACE measurements (Velicogna et al., 2020)) than climatological values. Defining an anomaly, however, requires defining a baseline period. For Greenland, we are able to extend the datasets back to 1850, and so a natural baseline would be the period of 1850–1900. This is the baseline used by the Intergovernmental Panel on Climate Change (IPCC) for surface temperature records, and is a relatively stable time period climatically over which the Greenland Ice Sheet was neither gaining nor losing significant mass (see next section). This also aligns reasonably with the pre-industrial control (*piControl*) runs that are part of CMIP.

However, for Antarctica, we do not have sufficient observational data to estimate the 19th century baseline flow rates (though evidence suggests that snowfall hasn't changed much, and the ice sheet was broadly in balance for several thousand years before the 1940s (Thomas et al., 2017; Steig et al., 2013)). Therefore, we are forced to define the anomaly relative to more recent conditions. Specifically, we compute the anomaly as the sum of grounded ice mass anomaly, shelf calving anomaly, and subshelf melt anomaly. The grounded ice mass anomaly is from the GRACE/GRACE-FO estimates (Döhne et al., 2023; Groh and Horwath, 2021) relative to 2002. The calving anomaly is provided by Davison et al. (2023) relative to 1997. The sub-shelf melt anomaly comes from one or the average of Davison et al. (2023) and Paolo et al. (2024) when they overlap, after setting the baseline to 1997. For modelers using the anomaly products, there is an implicit (and possibly incorrect) assumption that the ice sheets were close to balance over the baseline period. We will return to this assumption in section 3.2.

The products described below capture most, but not all, freshwater from the ice sheets, and make some simplifications. Specifically, Antarctic calving is limited to ice shelf calving and neglects non-shelf calving from smaller glaciers, and subshelf melting ignores frontal melt, although frontal melt is then implicitly included in the calving term. We also do not include basal melting of grounded ice for either ice sheet. Both the Antarctic calving and Greenland discharge terms include ice shelf

retreat, but we do not include ice shelf grounding line change for either ice sheet. These are assumed to be small compared to the resolved flows.

For both an overview and detailed discussion of all mass flows in Greenland and Antarctica, with a focus on observations, relative scales of terms, and how those mass flows relate to freshwater, see Mankoff et al. (2025b).

## 2 Modeling Needs

CMIP-class models are used to simulate many different numerical experiments. Usually, a pre-industrial control is run with quasi-mid-19th century conditions, followed by a historical simulation that starts from a point in the pre-industrial control and runs forward to the near-present using estimates of time-varying natural and human climate drivers. The historical simulation is extended into the future using scenarios based on consistent storylines of plausible future changes in the climate drivers. Other configurations extend back over the last millennium or deeper time, or are idealized in some fashion for easier comparison across models. Additionally, many groups use their ocean modules for ocean-only configurations (e.g., Danabasoglu et al. (2016)). The implications of adding a new forcing dataset needs to be considered for each of these different configurations and experiments, and this is explicitly done in Sections 4 and 5.

Conventionally, CMIP-class models assume that the climate was in quasi-equilibrium in the pre-industrial period. In reality, neither anthropogenic forcings nor slow responses to natural Holocene changes were zero during this period, though in practice this has not been a major issue. A bigger issue is the potential for 'climate drift' in the simulations due to the slow response of the deep ocean to surface conditions which implies that there can be sea level, carbon and energy imbalances for hundreds to thousands of years before the oceans reach equilibrium. These drifts can exist in any coupled climate model simulation, but are exacerbated by any non-conservation issues in the models. Non-conservation of mass or energy can arise through coding errors, but also through coupling of modules with inconsistent energy or mass formalisms, or through deliberate modeling choices. Examples range from the tiny (e.g., exogenous input of water through the oxidation of methane in the stratosphere that may not be matched by a sink,  $\approx 0.5$  Gt/yr) to the more significant (e.g., an assumption of no iceberg calving or freshwater flow from the ice sheets, \$\approx\$3300 Gt/yr from Antarctica (Depoorter et al., 2013), with approximately 500 Gt/yr of discharge from Greenland (Mankoff et al., 2020b)). Historically, there have been examples of energy non-conservation associated with sea ice-ocean coupling (Bitz and Lipscomb, 1999), and in ongoing efforts to couple ice sheets to climate models (Smith et al., 2021). Criteria for deciding whether a simulation is 'close enough' to equilibrium (such as a Top-Of-Atmosphere (TOA) radiative imbalance of  $

**Figure 2.** Region names and numbers used in data products for each ice sheet. The Greenland regions from Mouginot and Rignot (2019) and Antarctic regions from The IMBIE team (2018). Product provided as part of this work in GeoPackage (vector) and NetCDF (raster) format, including zones flood-filled into the surrounding ocean, available at https://doi.org/10.5281/zenodo.14020895.

statistical downscaling technique based on the local vertical runoff gradient applied to sub-grid topography (Fettweis et al., 2020). Prior to downscaling, MAR ran at 7.5 km resolution with forcing from the ECMWF Reanalysis project (ERA5). Runoff is assumed to route instantaneously and subglacially to the hydrologically-connected outlet (e.g., Chandler et al., 2021) at the ice/water or land/water interface (Mankoff et al., 2020a). A validation by Mankoff et al. (2020a) with all available individual stream gauges shows that on an annual average, runoff has an uncertainty of plus-or-minus a factor of two, or + 100 % / - 50 %. Averaging over larger areas (fjords, basins or over a larger regional scale) reduces the uncertainty to that of the RCMs – around  $\pm 15$  %.

220

The other significant contribution from the ice sheet freshwater mass flow comes from solid ice discharge across the grounding line of marine-terminating glaciers (Fig. 4). Annual values for all of Greenland at present are  $\sim$ 475 Gt yr<sup>-1</sup>. Ice sheet grounding line discharge is computed using surface velocity, ice thickness, and ice density (Mankoff et al., 2020c) at flux gates  $\sim$ 5 km upstream from the grounding line. As ice crosses the grounding line, it is lost to the ocean via frontal ablation, which consists of iceberg calving and submarine melt. The partition between these two processes is highly uncertain, varies in space and time, and there is no agreed upon proportioning, despite agreement that warm (and warming) oceans play an important role (Wood et al., 2021). Summer 2008 field measurements and related work in central west Greenland informed an estimate that 20–80 % of summer ice-front fluxes are directly melted by the ocean, with lower values expected in wintertime (Rignot et al., 2010). Recent efforts to separate solid discharge from the sum of submarine melt/subaerial melt and sublimation on decadal

**Figure 3.** Time series of Greenland runoff split by source (ice, land) and destination - runoff then enters the fjord surface, or enters the fjord at depth via subglacial discharge from marine terminating glaciers. We note that land-sourced runoff does re-enter the subglacial system and discharge into a fjord at depth. Product provided as part of this work available at https://doi.org/10.5281/zenodo.14020895.

scales indicated 90 % solid discharge and 10 % melt/sublimation (Kochtitzky et al., 2023). Fieldwork on a large floating ice tongue in north Greenland in 1992 suggested that as much as 75 % of the mass loss was via submarine melt (Rignot, 1996). However, many extended ice shelves around Greenland have since collapsed. Furthermore, discharge also has a  $\sim$ 10 % uncertainty due to uncertainty in ice thickness at the flux gates (Mankoff et al., 2020c). Here, we use the 20 – 80 % estimate mean and spread and assume a 50 %  $\pm$  30 % submarine melt vs. calving estimate, but flag the high uncertainty of this assumption and likely heterogeneity of values across the ice sheet, and stress the need for further detailed studies separating contributions from calving and submarine melt.

235

240

245

250

Finally, the  $\sim$ 5 km distance between the gates and the grounding line is also often equivalent to several years or more of time due to ice flow speeds, and our estimates do not take into account mass loss below the flux gate due to surface mass balance changes (e.g., surface melting). This is likely approximately 25 Gt yr<sup>-1</sup> (Kochtitzky et al., 2023). However, we do explicitly include frontal advance and retreat of the terminus. This has added  $\sim$ 1000 Gt between 2002 and 2022 (Greene et al., 2024), or  $\sim$ 50 Gt yr<sup>-1</sup> or  $\sim$ 10 % to the annual discharge reported by Mankoff et al. (2020c).

Additional sources of freshwater include frontal melt expressed as grounding line retreat which is included in these products, and basal melt of grounded ice which is not included. Basal melt, from geothermal heat flux, frictional heat from sliding, and viscous dissipation in the turbulent subglacial flow adds an additional  $\sim$ 20 Gt yr<sup>-1</sup> (Karlsson et al., 2021). We neglect the basal melt of grounded ice because it is both one of the smallest terms Mankoff et al. (2025b) and it is mostly due to steady state processes, and therefore has no anomaly, which is the focus of many of the products presented here.

The energy required for surface and basal melt runoff does not come from the ocean and so this water enters fjords or the open ocean in the liquid phase. By contrast, submarine melt occurs through the extraction of oceanic heat at the ice/ocean interface, even if this occurs in small fjords not included in the models due to resolution. Icebergs melt at varying distances

**Figure 4.** Time series of Greenland a) ice discharge, and b) ice discharge anomaly (w.r.t. 1850–1900) (right) from flux-gates ∼5 km upstream of the terminus. This discharge is divided, roughly equally, into submarine melt and iceberg calving. Data post-1986 are from regionally-defined observations. Pre-1986, the data is derived from the total ice sheet changes, regionally partitioned based on the average of the first five years of regionally-defined observations. Product provided as part of this work available at https://doi.org/10.5281/zenodo.14020895.

from their source (Fig. 5), and the energy for that is predominantly from the upper ocean (Savage, 2001; Martin and Adcroft, 2010). This is energetically equivalent to adding the freshwater as ice into the ocean.

Additional sources of FW into the Arctic and sub-polar North Atlantic originate from other land ice areas across the Arctic. More than half of glacier and ice cap volume (that is non-ice sheet land ice) lies in the Arctic, primarily in the Canadian Archipelago, Svalbard, Russian Arctic, Iceland and peripheral glaciers and ice caps around Greenland. The mean FW input from all of these sources has been about a third of that from the GrIS and since 1980 the anomalous FW amount has also been about a third of that from the main ice sheet (Igneczi and Bamber, 2025).

Finally, we provide estimates of the mass imbalance (relative to 1850–1900), defined as SMB anomaly minus discharge anomaly (Fig. 6). However, mass changes are only somewhat related to freshwater changes, and when working in anomaly space the relationship even less clear. As one example, a region that increases mass loss, may do so due to a decrease in runoff and calving offset by an even larger decrease in snowfall inputs. In this scenario, total freshwater outputs (solid and liquid) decrease, but the region is losing mass relative to years when there were larger snowfall inputs and larger runoff and calving outputs.

#### 3.2 Antarctica

270

The FW fluxes from Antarctica are substantially different from those in Greenland (c.f., Mankoff et al., 2025b). Runoff is  $\sim$ 435 Gt yr<sup>-1</sup> in Greenland and  $\sim$ 10 Gt yr<sup>-1</sup> in Antarctica (Fettweis et al., 2020; Kittel et al., 2021), while sub-shelf melting is  $\sim$ 25 Gt yr<sup>-1</sup> in Greenland and  $\sim$ 1000 Gt yr<sup>-1</sup> in Antarctica (Wang et al., 2024, this work). Positive SMB is balanced by discharge: the majority (approximately 90 %) enters ice shelves, which release freshwater to the ocean approximately equally

**Figure 5.** Maps showing the annual average iceberg melt rates as a function of their source region in Greenland. Similar maps with monthly resolution are also available. Each map multiplied by cell area and summed equals one, so that these can be used for distributing freshwater inputs computed elsewhere. See Fig. 2 for region names and numbers. Product provided as part of this work available at https://doi.org/10.5281/zenodo.14020895.

Figure 6. Greenland mass anomaly per region. Product provided as part of this work available at https://doi.org/10.5281/zenodo.14020895.

**Figure 7.** Maps showing the annual average weighting function for iceberg melt rates as a function of their source region in Antarctica. Similar maps with monthly resolution are also available. Each map multiplied by cell area and summed equals one, so that these can be used for distributing freshwater inputs computed elsewhere. See Fig. 2 for region names and numbers. Product provided as part of this work available at https://doi.org/10.5281/zenodo.14020895.

through calving and sub-shelf melt (Greene et al., 2022). The areal distribution of iceberg melt is also more extensive than around Greenland, and the nature of the icebergs themselves (tabular) is also different (Fig. 7).

The Antarctic SMB is best estimated from well-calibrated regional climate models driven by atmospheric reanalyses, and mostly consists of snow accumulation, with a relatively small negative contribution from surface and blowing-snow sublimation, and negligible contributions of rainfall and runoff (Agosta et al., 2019; Gadde and van de Berg, 2024). No significant trends in Antarctic-wide SMB since 1979 have been identified (Mottram et al., 2021).

According to multiple estimates from satellite observations of temporal changes in ice sheet flow, ice sheet volume, and Earth's gravity field, the grounded ice sheet has been losing mass since satellite records began in the early 1990s (Otosaka et al., 2023a), mostly due to increased ice discharge across the grounding line (Rignot et al., 2019) (see Fig. 8). Ice shelves have also lost mass since the 1990s, as the mass gained through increased grounding line discharge has been overwhelmed by increased ice loss through calving and basal melting (Davison et al., 2023; Rignot et al., 2013; Slater et al., 2021) (see Fig. 9).

280

We use the input/output method (mass balance = SMB - discharge) to estimate grounded mass loss (Rignot et al., 2019), a similar method to estimate ice shelf basal melt (SMB inputs minus elevation change; Davison et al. (2023); Paolo et al. (2024)), and remote sensing image time series of iceberg calving (Davison et al., 2023).

**Figure 8.** Antarctica grounded mass change (left) and freshwater anomaly (right). Left: Light gray lines are the 18 regions (Fig. 2). Thick black line is sum of all regions, and dashed line is cumulative sum. Right: Components contributing to freshwater anomaly without lag applied to calving. We recommend spreading Antarctic calving over 10 years as iceberg melt is not instantaneous. Product provided as part of this work available at https://doi.org/10.5281/zenodo.14020895.

**Figure 9.** Antarctica sub-shelf melt and calving mass flow rates, baseline, and anomalies. We recommend spreading Antarctic calving over 10 years as iceberg melt is not instantaneous. Product provided as part of this work available at https://doi.org/10.5281/zenodo.14020895.

The aforementioned methods provide good estimates of Antarctic freshwater mass flow rates into the Southern Ocean since 1997, but more assumptions are needed to estimate freshwater between 1850 and 1996. In the reconstructions from Frederikse et al. (2020), the Antarctic contribution to sea level rise was low over 1900–1980, with 0.06±0.06 mm yr<sup>-1</sup>, though it seems that Amundsen Sea Embayment (ASE) grounding line retreat began in the 1940s (Smith et al., 2016; Clark et al., 2024). However, it is beyond the scope of this paper to reconstruct the changes in the grounded ice sheet prior to the 1990s, and so we assume a quasi-steady state back to 1850. As new reconstructions become available, we hope to be able to revisit this assumption in future work.

A combination of firn core analysis and regional climate modeling indicates that the Antarctic SMB was 5 % smaller over 1850–1980 than over 1990–2010 (Thomas et al., 2017). Similar results were obtained from an emulation of a regional climate model constrained by multiple CMIP6 models since 1850 (Jourdain et al., 2025). From all this and from present-day estimates, it is possible to estimate the ice discharge across the grounding line that was needed to maintain a steady grounded ice sheet over 1850–1980. Potentially, one could refine the calculation for the ASE by using knowledge of the grounding line position in the 1940s for Pine Island and Thwaites glaciers (Smith et al., 2016; Clark et al., 2024) and calculate the volume of ice lost to shift from past geometries to present geometries, but that is beyond the scope of this paper.

Since ice shelves buttress the grounded ice sheet, a grounded ice sheet in steady state likely goes together with ice shelves in steady state. Under this assumption, we assume an ice-shelf mass balance at the beginning of the well-observed period, use the steady ice discharge across the grounding line derived from SMB above, and estimate the fluxes through calving and basal melting required to have a net zero ice shelf mass loss (steady state), though this could be an underestimate in areas (such as the Amundsen Sea, where retreat is ongoing).

While the net loss of grounded ice in recent decades can be constrained by the GRACE and GRACE-FO measurements (Bamber et al., 2018b), the uncertainties on individual basin discharge will inevitably be higher. Nonetheless we feel it is important to retain some of the spatial information from the IMBIE analyses (Otosaka et al., 2023a) so that we can capture potentially varying trajectories in the past (and possible future) between the Peninsula, West and East Antarctic Ice Sheets (WAIS and EAIS).

#### 3.3 Summary of data products

We provide the following data products. All time series are annual temporal resolution and units Gt yr<sup>-1</sup>. All products have regional resolution per ice sheet (Fig. 2). The steady state iceberg melt maps are at 0.5 degree spatial resolution and monthly temporal resolution. Units for iceberg melt maps are m<sup>-2</sup>, and when multiplied by any specific model's cell area maps should be adjusted to sum to one. This set of inputs is substantially more complex than previous specifications (such as described by the SOFIA initiative (Swart et al., 2023)), but can be simplified or consolidated as needed.

Greenland runoff from 1950 through 2023 at monthly resolution and by region (Fig. 3). The source data (Mankoff, 2020) has daily temporal and stream spatial resolution (Mankoff et al., 2020a) but is here resampled to monthly and regional resolutions. This product includes four variables that split the runoff by its two sources (ice sheet or peripheral land) and two

destinations (fjord surface via sub-aerial stream or subglacial discharge at the bottom of marine-terminating glaciers). We assume climatological values prior to 1950.

Greenland discharge from 1840 through 2023 at annual resolution and by region (Mouginot and Rignot, 2019). The source data (Mankoff and Solgaard, 2020) has Greenland-wide spatial resolution from 1840 through 1985 and regional spatial resolution from 1986 onward (Mankoff et al., 2020c). To provide regional resolution for the entire time series we take the average of the earliest five years of regional resolution (1986–1990) to determine the relative contribution of each region to the whole, and then split the whole by that proportion from 1840 through 1985 (Fig. 4). We provide Greenland discharge and a protocol recommendation for the separation between calving and submarine melt, but not a data product for this separation or these terms, because this is dependent on how and whether models resolve fjords.

**Greenland mass anomaly**. We provide estimates of the mass imbalance (relative to 1850–1900), defined from the change in modeled SMB minus the change in discharge which, by construction, will average to zero over the baseline period for each region (and for the ice sheet as a whole). Additionally, we provide the separate discharge and runoff anomalies (Fig. 6).

Antarctic calving from 1997 through 2021 at annual resolution by region. This is derived from Davison (2023) with the only modification being aggregation by region.

Antarctic submarine melt from 1991 through 2021 at annual resolution and by region (Fig. 2). The source data comes from both Davison (2023) and Paolo et al. (2024). Where the Davison et al. (2023) and Paolo et al. (2024) time series overlap we take the mean of the two after aggregating by region.

**Antarctic freshwater anomaly** is provided relative to quasi-1990 conditions. This is defined as the modeled SMB minus the calving and submarine melt terms, normalized so that it is zero in 1990.

**Iceberg melt maps** are provided for both the areas surrounding Greenland and Antarctica. These maps were generated for this work (see supplemental code). We provide steady state annual mean normalized maps (units are  $m^{-2}$  and maps should sum to 1 when multiplied by cell area) at 0.5 ° longitude by latitude spatial resolution. Input for Greenland comes from Marson et al. (2024) and for Antarctica comes from Olivé Abelló et al. (2025); Mathiot and Jourdain (2023).

#### 3.4 Implications for modeling of sea level

Freshwater input from the grounded ice sheets is a significant component in present day sea level rise (Dieng et al., 2017) (along with mountain glacier melt, ocean warming, groundwater, etc.). Freshwater from floating ice sources has a much more muted impact because of the hydrostatic compensation, but has an influence through halosteric and thermosteric effects (Jenkins and Holland, 2007; Noerdlinger and Brower, 2007). Both terms however influence ocean circulation and stratification equally.

CMIP-class models have not generally been used to calculate global sea level rise since they do not have a complete accounting of all the terms. However, they have been used as input into more comprehensive assessments (e.g., Kopp et al., 2023) through their estimates of ocean (thermo)steric effects and changes in ocean dynamic topography. The use of the freshwater terms presented here can help improve the representation of sea level change in CMIP7 in a number of ways, though there are significant caveats.

First, if an ocean module is volume conserving and uses equivalent salinity fluxes to represent freshwater fluxes, only the (small) halosteric effect of the freshwater input will be included. Ocean modules that are mass conserving and have natural boundary fluxes for freshwater will additionally represent a barysteric effect (roughly 2.8 mm of sea level rise per 1000 Gt of freshwater input). However, if the ocean module does not include the ocean cavities below the floating ice shelves, or assumes those cavities are rigid, or does not allow the mass of floating ice to impact the pressure in the ocean, then the hydrostatic compensation will not be represented and the total sea level rise in the model will be too high (roughly by the net amount of freshwater from floating ice times 2.8 mm/1000 Gt). Theoretically, modelers could compensate for this by removing a equivalent mass for the no-longer displaced ocean (i.e., for the net amount of freshwater addition from floating ice shelves, the same mass of (deeper) ocean water could be extracted from the system). The characteristics of this sea water is likely to represent (in the Southern Ocean) Circumpolar Deep Water that is being brought up and over the continental shelves. This would allow for the stratification/salinity changes to be represented, while the impact on sea level would be purely steric, but we are unaware of any group that has taken this into account.

## 4 Modeling approaches

We first discuss the choices available for the *piControl* simulations and then turn to the historical and future simulations. Most climate models already have code that allows for meltwater/iceberg discharge from ice sheets, and the easiest implementation across the control and historical simulations is to use the same coding framework but with adjusted, more-realistic forcing.

First, we assume that all ESMs have snow models over the land ice component that will calculate the SMB and any potential runoff. Most (if not all) models have a scheme that routes that runoff downstream. Some models aggregate the runoff globally and distribute it to the ocean, though we recommend that models without a routing scheme aggregate the total runoff by individual basins (see Appendix A, Fig. 2) and spread it at the surface of coastal points adjacent to the basin.

Some simplifications can be made that we judge are unlikely to have a large impact in CMIP-class climate models in most cases. For instance, we assume that spatial distributions of freshwater forcing are fixed in time, and that the depth over which the mass flows are applied is also fixed both in space and time. Note that depth profiles for freshwater forcing at the fjord mouth or calving front might be different from the depth profile associated with melting icebergs (Savage, 2001). These are reasonable approximations today, but may become less valid in much warmer climates (such as extended SSP5-8.5 simulations) (Siahaan et al., 2022; Coulon et al., 2024).

Additionally, there are notable seasonal cycles in runoff, ice discharge, and in iceberg drifts and melt distributions, as well as in fjord stratification (Jackson and Straneo, 2016; Merino et al., 2016; Bamber et al., 2018b). For completeness, we provide variations in the iceberg melt by month, as well as annually. Some recent work suggests that there may be a small sensitivity in the Southern Ocean to the seasonality of the iceberg melt (Kaufman et al., 2024), but exploring this more deeply is beyond the scope of this paper. Overall, we generally assume that we can neglect the seasonality of discharge.

We should be clear that there is no perfect solution. Assuming (realistically) that all models will have biases in the SMB over the ice sheets, it is impossible to simultaneously satisfy mass conservation and have the correct runoff and discharge

rates. Different approaches effectively prioritize different aspects, and that is a judgement call that needs to be made by each modeling group. We start the discussion with an assessment of what is needed for any simulation, followed by a focus on the pre-industrial control runs, and what that implies for other subsequent simulations.

#### 4.1 General considerations

There are a number of choices that can be made relating to the temporal pattern of the freshwater additions, the spatial pattern of the addition, the phase (or energy) associated with the addition, and the degree to which the addition is regionally resolved. These choices should be influenced by the degree of complexity the developers envisage for the historical freshwater flows, i.e., if regionally defined iceberg melt changes are wanted in the historical simulations, it is most consistent for them to be included in all other simulations also. Additionally, whether iceberg melt, glacier frontal and basal melt, and surface and subglacial runoff are dealt with separately can also differ. We appreciate that there will be differing appetites for additional work to implement this, and thus we provide guidance and forcing for a range of approaches that are progressively more complicated, but that yield the same global (and hemispheric) averages.

#### 4.1.1 Timescale

The implied freshwater losses/gains can be made at each time step, accumulated over a month or a year or longer, and used to update the additions instantaneously, or once a month, once a year, etc., or with a relaxation time. A very short timescale (less than a year) would affect the seasonal cycle of freshwater flow, but a very long relaxation time (greater than decades) would proportionately increase the time to reach equilibrium. For example, the GISS-E2.1 and IPSL-CM models accumulate the implicit freshwater mass and energy accumulation on an annual basis and distribute it with a 10-year relaxation constant to minimize excessive interannual variability (Kelley et al., 2020; Boucher et al., 2020), while CanESM moves the implicit freshwater mass and energy to the column liquid runoff, which then follows the river routing scheme to the continental edges, without additional relaxation time than the river flow time (similarly as described in Arora et al., 2025).

#### 405 4.1.2 Spatial distribution of prescribed freshwater

Where the freshwater from the ice sheets/ice shelves enters the ocean needs to be prescribed or parameterized. There are two main components: water that enters the ocean model domain locally to the ice sheet/shelf and water associated with icebergs that might leave the local area. For the latter flux, models that do not resolve icebergs interactively must prescribe the spatial distribution of the meltwater from icebergs. This distribution can be global, hemispheric, a single map for each ice sheet, or separate maps for each (major) drainage basin (Fig. 2), and a few additional areas included for other locations with tidewater glaciers (Alaska, Iceland and Svalbard). At the present time, it is challenging to attempt to implement regional-scale timeseries that are resolved to a finer spatial scale than these basins, but that might change once more robust coupled ESM/ISMs are available.

In Greenland, ice discharge is computed as solid ice through flux gates  $\sim$ 5 km upstream of the terminus because estimating ice thickness directly at the terminus introduces more error than the error introduced via SMB changes downstream of the flux gates (Mankoff et al., 2020c). That ice flux is divided into submarine melt (primarily frontal melt) and iceberg calving at the terminus. We apply an estimate for this division of  $50\% \pm 30\%$  (Rignot et al., 2010) but reiterate that the partitioning is highly uncertain and spatiotemporally variable. Frontal melt, typically ignored in Antarctica (but implicitly included in that calving product), is a major source of freshwater in Greenland.

For models with iceberg representations, modelers can decide how to distribute the total calving anomaly across the considered iceberg classes in their models (Ackermann et al., 2024). This could require separate iceberg size distributions for every IMBIE basin, including the calving of giant tabular icebergs if supported by the model, as the distributions can differ strongly between different calving sites (Wesche et al., 2013) and with distance from the calving front (Kirkham et al., 2017). Alternatively, iceberg sizes could be initialized following a single power-law distribution, e.g., with slope  $-1.52 \pm 0.32$  (Tournadre et al., 2016) or  $-1.77 \pm 0.04$  as determined for near-coastal regions of Antarctica (Barbat et al., 2019, their Fig. 5).

For most CMIP-class models, such as the IPSL-CM model (Boucher et al., 2020), maps of iceberg melt can be assigned to a single or multiple basins per ice sheet based on, for instance, the melt pattern obtained by Merino et al. (2016) in a 0.25° global ocean simulation with Lagrangian icebergs. Other available melt patterns, including the effect of Lagrangian giant icebergs, could be used as well (Rackow et al., 2017b; Bi et al., 2020; Marsh et al., 2015), but it is an open question how to average over the effect of individual, rare giant iceberg trajectories that calve on decadal timescales (Stern et al., 2016; Rackow et al., 2017a). More sophisticated methods are also being developed (e.g., Sulak et al., 2017; Shankar, 2022).

The vertical and regional distribution of the freshwater is a function of many small-scale processes and local ocean circulation. For instance, for marine-terminating outlet glaciers in Greenland and the West Antarctic Peninsula, subglacial discharge emerging at the grounding line during summer months can drive substantial plume-driven upwelling. Furthermore, these plumes often equilibrate well below the ocean surface and entrain large amounts of seawater as they rise, diluting the meltwater signal and increasing the plume volume (Beaird et al., 2018). For example, Slater et al. (2022) estimates summer subglacial discharge from 136 tidewater glaciers in Greenland, with flows of 0.02 Sv (630 Gt yr<sup>-1</sup>) of freshwater at the grounding line. However, entrainment from rising plumes drives an upwelling of 1.07 Sv ( $\approx$ 34000 Gt yr<sup>-1</sup>), approximately 50 times greater than the original subglacial discharge. Furthermore, the outflowing plume-modified freshwater equilibrates primarily at 25–200 m depth. To account for these processes, a glacier-resolved plume product could be used to force subglacial discharge plumes along the coastal periphery of Greenland (e.g., Slater et al., 2022).

Additionally, localized submarine melt along the width of the glacier terminus, and the resultant fjord-scale circulation driven by the combination of meltwater and subglacial discharge, can drive substantial modification in freshwater forcing from outlet glaciers (Carroll et al., 2017; Davison et al., 2022). Within the ocean module, the representation of coastal bathymetry and land mask, and how well they resolve fine-scale bathymetric and coastal features, i.e., fjords and bays, will also dictate how this freshwater should be accommodated. If fjords are included with sufficient width to resolve fjord-scale circulation, the freshwater forcing could be implemented close to the glacier terminus. For coarser-resolution models that do not resolve fjords (e.g., most CMIP-class models), a transfer function or estuarine box model (e.g., Sun et al., 2017, 2019), which could

be run offline, may be needed to account for fjord-scale mixing of the freshwater signal before it reaches the shelf. In lieu of that, the freshwater might be entered over a representative depth near the fjord mouth (Straneo and Cenedese, 2015). In Greenland, meltwater routed via subglacial discharge conduits is distributed throughout the top  $\sim$ 250 m with a peak at  $\sim$ 100 m (Slater et al., 2022, Fig. 3c). However, the effect of representing plume-driven upwelling and mixing on regional ocean conditions has not been assessed. Although recent advances in fjord box models (e.g., Slater et al., 2025) provide a promising avenue for representing these processes in CMIP-class models in future, for now we recommend distributing all freshwater from Greenland icebergs, submarine melt, and subglacial discharge evenly in the top 200 m. This assumption will capture the overall extraction of energy from the ocean by melting ice, but may result in overestimation of near-surface cooling.

For Greenland, iceberg melt distributions are also poorly known on the regional to pan-Greenland scales, but can be constrained regarding an upper depth limit by glacier ice thickness (analogous to Antarctica). Both iceberg melt models (Moon et al., 2017) and differencing of satellite data-derived digital elevation models (Enderlin et al., 2018) provide some basis for understanding iceberg melt distributions, but the former is so far limited to a single fjord application and the latter is limited regarding vertical freshwater distribution.

We use outputs of a dynamic and thermodynamic iceberg model included in the Nucleus for European Modelling of the Ocean (NEMO v3.6; Gurvan et al. (2017)), with modifications that allow thick and concentrated sea ice to lock icebergs within it (Marsh et al., 2015; Marson et al., 2024; Rackow et al., 2017a). This iceberg model is forced with the full solid discharge rates from Bamber et al. (2018a) and generates Lagrangian particles containing icebergs of ten possible size classes, ranging from 60 m to 2,200 m in length (Martin and Adcroft, 2010). Model outputs include individual particle trajectories and associated information about the icebergs' mass at every model day. These trajectory files allow us to connect the initial particle location to the nearest of the seven IMBIE regions in Greenland, and then track meltwater (estimated from iceberg mass loss) spatially for the life of each iceberg. This generates the meltwater maps shown in Fig. 5 and in the supplemental data. It is worth noting that the iceberg model does not yet include the "footloose" parameterization (Wagner et al., 2014), which means that icebergs in the model break up more slowly than observed. The lack of representation of this deterioration mechanism could contribute to a broader iceberg distribution compared to reality (Huth et al., 2022), though there are other issues that can arise in such simulations (Wagner and Eisenman, 2017).

Our iceberg spatial melt maps (Fig. 5) do not resolve fjords (similar to most CMIP models). As discussed above, Rignot et al. (2010) estimate that 50 % of discharge is melted at the ice front within a fjord, and we assume that 50 % of icebergs melt within the fjord and the remaining ice melts non-locally following the iceberg melt maps. This implies that, if a fjord is resolved, 50 % of the discharge term should be distributed as submarine melt within the fjord, or if a fjord is not resolved 75 % of the discharge should be added as submarine melt at the fjord-adjacent grid cells. In either case, 25 % of the discharge should be assumed to go to far-field icebergs. Note that when using these iceberg distribution maps, they will need to be adjusted for a new model or land-ocean mask. It is straightforward to apply the ocean mask, and then reweigh the distribution so that the sum of the melt distribution should equal one.

For Antarctica, the iceberg meltwater should be distributed according to the spatial melt maps (Fig. 7) and the sub-shelf meltwater should be horizontally distributed along the front of unresolved ice-shelf cavities. The iceberg melt maps were

obtained from the Nucleus for European Modelling of the Ocean (NEMO v4.2; Gurvan et al. (2022)) and its Lagrangian iceberg module (Marsh et al., 2015; Merino et al., 2016). A 0.25° model configuration of the Southern Ocean was used, forced by a normal year from the JRA55 reanalysis, and the calving of iceberg particles fed by the observational calving flux of Rignot et al. (2013) and distributed into 10 iceberg size classes of up to 3.6 km<sup>2</sup>. The melt patterns were saved separately according to their calving location in the 18 Antarctic IMBIE regions (Otosaka et al., 2023a). For more details on the model configuration and its evaluation, see the "NoGr" configuration in Olivé Abelló et al. (2025). For the horizontal distribution of sub-shelf meltwater, guidelines are provided in the data repository to identify the front regions of the unresolved cavities.

Iceberg thickness and the depth at which their meltwater is injected are poorly known, though the depth of iceberg and ice-shelf bases provide a good constraint on the maximum depth at which freshwater is injected. The upper limit for iceberg thickness is the ice-shelf thickness at its calving front (based on BedMachine v3, Morlighem (2022)). However, small icebergs are thinner than the ice-shelf calving front, and the iceberg thickness reduces as they melt. In the absence of accurate spatially and temporally varying data, we suggest spreading the iceberg meltwater uniformly over the upper 200 m. Naughten et al. (2022) distribute the iceberg meltwater over the upper 350 m in the Amundsen Sea, but this is probably specific to that region where numerous thick icebergs are calved (Olivé Abelló et al., 2025).

For the vertical distribution of the Antarctic sub-shelf meltwater, models that do not resolve sub-shelf ocean cavities should ideally distribute the ice-shelf meltwater between the depth of the deepest part of the cavity (usually near the grounding line) and the minimum ice-shelf depth at the front, for individual ice shelves or drainage basins (Mathiot et al., 2017). The corresponding maximum and minimum depths of sub-shelf melt injection provided in the data repository were inferred for each of the 18 Antarctic IMBIE regions from ice-shelf draft and bathymetry observational estimates at 2 km resolution (based on BedMachine v3, Morlighem (2022)), for each drainage basin separately. A simpler but less accurate method is to distribute it uniformly along the Antarctic coastline and between 203 m depth and 534 m depth, which are the Antarctic-averaged values of the aforementioned depths.

## 4.1.3 Energy fluxes associated with the freshwater

The phase of the freshwater, or more precisely, the energy associated with the mass flow, is an issue with potentially important consequences. Since the SMB anomaly over the ice sheets is due to the accumulation of snow (a "negative latent heat") and has "negative" sensible heat (assuming that the Energy Reference Level (ERL) is liquid water at 0 °C), that same (negative) energy needs to pass into the ocean at steady state. However, not all the energy required to melt the discharge comes from the ocean. There is a very small energy flux from the atmosphere to the protruding parts of icebergs, and there is also possibly warming and/or melting within the ice sheets driven by geothermal heating, strain, basal friction or through release of the potential energy of the snow that fell at altitude. These terms are nonetheless small compared to the sub-shelf melting, in-fjord melting and sub-surface iceberg melting, all of which draw energy from the ocean. Thus, to first approximation, all freshwater additions to the ocean can be considered as ice i.e., the latent heat consumption of melting ice should be included alongside the corresponding freshwater flux. The additional sensible heat contribution from adding ice at a nominal -20 °C and then warming

the melted water to the ambient temperature is up to 15 % of the latent heat; this can have impacts on the surrounding ocean temperature, density, overturning, and sea ice formation.

If there is a desire to be more faithful to the oceanography, sub-shelf melt could be added in as liquid at the pressure melting point, which becomes colder with increasing depth below the surface. This would imply a small energy imbalance and could be thought of as an implicit change to the geothermal flux. We would not suggest doing this for submarine melt for meltwater at the ice front or within a fjord, because even if the cavity or the fjord are not resolved by the ocean model, the source of energy for the melting is the ocean.

#### 4.1.4 Tracers

Some modellers may wish to include some tracers along with the freshwater, such as the isotopic content ( $\delta^{18}O$  or  $\delta D$  (e.g., Brady et al., 2019)), mineral dust, dissolved  $CO_2$ , iron, or nutrients (Hawkings et al., 2015), but the details of these examples are beyond the scope of this paper. A zeroth-order estimate would be to assume a constant but representative tracer concentration value for the tracers in the climatological and anomalous freshwater flux, ideally derived from observations. If conservation between the SMB and discharge is required, the mean tracer concentration could be calculated from the overall SMB tracer budget.

## 4.2 Modeling approaches in pre-industrial controls

As discussed above, the freshwater mass, tracer and energy losses/gains from the resolved components in the pre-industrial simulations need to be added back in to the ocean to allow for an eventual quasi-equilibrium. This requires that diagnostics of the SMB on the same regional basis as for the freshwater inputs to the ocean.

## 535 4.3 Modeling approaches in historical simulations

For the models that calculate and apply a pre-industrial (PI)-control freshwater mass flow (as described above), two different approaches have been used for historical simulations (see Fig. 10, top panel): the first approach is to fix discharge at pre-industrial levels (Type 1 approach), while the second approach assumes continuing ice sheet mass balance (Type 2 approach) and thus updates the (regional) discharge as a function of changes in net (regional) SMB. The updating in this latter case can vary in effective relaxation time as described above. The choice of approach also applies for scenario or more idealized simulations.

In historical simulations to the early 21st century, the net SMB in Antarctica generally becomes more positive (increases in snow accumulation outweigh sublimation and runoff changes) (Purich and England, 2023) while for Greenland it is the opposite (there is a greater increase in runoff than accumulation) (Hofer et al., 2020). Thus, for existing models taking the Type 1 approach (fixed discharge) there will be a net loss of water (and small energy gain) in the climate model from the net accumulation in Antarctica and a gain of mass (and energy loss) from Greenland. In models with a Type 2 approach (ice sheet mass balance), there would be an increase in Southern Ocean freshwater inputs, and a decrease in discharge in Greenland. For

example, in the GISS-E2.1-G model, Antarctic discharge increased by 2 % and Northern Hemisphere (Greenland) discharge decreased by 4 % in 1979–2014 compared to the pre-industrial (Miller et al., 2021). Given that both ice sheets have been losing grounded ice mass in recent decades, this implies neither approach, as currently operated, matches the sign of discharge changes in both regions and, even where the sign is correct, the magnitude of any change is (unsurprisingly) not close to observations.

Regardless of the approach, the datasets described above can be used to improve the match to the observed changes in the freshwater flows. For models using a Type 1 approach, we can add the anomalous discharge to the existing discharge amounts. This ensures that freshwater distribution change through time is reasonable, but the total net addition of water may not be correct depending on the accuracy of the SMB calculations. For models using a Type 2 approach, which assumes ice sheets in mass balance, it is straightforward to add the anomalous freshwater flow (discharge and runoff) to the updating discharge amount ensuring that the implied mass changes of the ice sheets and ice shelves are correct regardless of the SMB calculation. In each case errors (such as might exist) in the modeled SMB change will be implicitly associated with different reservoirs. For the Type 1 approach, an error in the SMB change will translate into an error in the implicit mass change of the ice sheets. For the Type 2 approach, such an error will be expressed through the discharge amounts, while preserving the implied ice sheet mass balance (see Fig. 10, bottom panel, for schematics of the proposed frameworks for both Type 1 and Type 2 approaches).

An additional consideration may be the seasonality of the changes in the discharge. The simplest assumption is to only update the freshwater inputs annually in line with the assumption discussed above of assuming that discharge doesn't vary much through the year.

Where models might resolve some of these components, e.g., by including the largest ice shelf cavities under the Ross or Filchner-Ronne ice shelves, or by resolving iceberg transport and melt, we suggest a partial use of the provided forcing.

#### 5 Recommendations

We briefly summarise our recommendations for specific model experiments, starting with the *piControl* and the *historical* simulations. We discuss below how other experiments requested by CMIP (broadly speaking) could be made consistent with the historical simulations, although we acknowledge that these efforts will require substantially more research.

#### 5.1 *piControl* simulations

- Runoff should be aggregated over each major basin and either routed to the ocean, or put into the relevant coastal grid boxes at the surface. However, we do acknowledge that approximately half of the runoff from Greenland (and perhaps increasingly from Antarctica, if surface melt of grounded ice becomes more extensive in a warming climate) is routed subglacially and enters the ocean at the depth of the grounding line.
- The implied discharge per basin should be estimated from the net SMB over each basin, and distributed 50:50 between iceberg calving and submarine melt (in the absence of fjords), or 25:75 for basins with major fjords. A 10-year relaxation timescale has proven useful in previous studies.

Figure 10. Schematics of the current (top) and proposed (bottom) approaches for the FW mass flow rates in climate models. In the schematic, F is the computed *picontrol* discharge,  $\Delta F$  is the change in the discharge (derived from observations), and the two approaches refer to the description in Section 4.2.

- The iceberg calving flux from each basin should be spread in the ocean according to the weightings per basin provided in the maps above,
- The sub-shelf melt flux should be spread uniformly over the ice-shelf cavity depths for all ice shelf fronts in that catchment.
- Iceberg fluxes and submarine melt (i.e., all discharge) should be injected as ice over a range of depths in the ocean. We
   suggest that some account be made of sensible heat, but this is a small term (< 15 %).</li>

Various simplifications are possible: the net SMB can be aggregated for the whole ice sheet, and the iceberg melt distribution can be taken for the whole ice sheet, ignoring the need to do separate calculations per basin. Also various complications can also be incorporated depending on whether some ice-shelf cavities or fjords are included in the ocean module domain.

#### 5.2 *historical* simulations

600

605

- Modelers need to decide whether they want a Type 1 or Type 2 approach to the discharge. For models with a Type 1 approach (an initial assumption of constant discharge), anomalous discharge amounts need to be added to the *piControl* discharge. For models with a Type 2 approach (with an initial assumption of continuing ice sheet mass balance), modelers need to add the anomalous FW flows to the calculated discharge.
  - Maps, depths, and partitions for the iceberg-related and local fluxes will be the same as for the *piControl*.

## 595 5.3 Idealized simulations

As part of the CMIP DECK runs, groups are often asked to submit multiple idealized scenarios (such as 1 % increasing  $CO_2$ , abrupt  $4 \times CO_2$ , flat10 (fixed 10GtC/yr emissions), etc.) that are subsequently used to characterize metrics such as the Equilibrium or Effective Climate Sensitivity (ECS), Transient Climate Response (TCR) or the Transient Climate Response to (cumulative carbon) Emissions (TCRE). Climate models that have interactive ice sheets (and thus the ability to calculate the changes in the freshwater forcing) will likely have different ECS, TCR and TCRE than models with either Type 1 or Type 2 approaches for the implicit ice sheets. It is therefore worth thinking about whether there are usable protocols for non-ISM simulations, that would give a more coherent response.

Increasing meltwater can act as a negative feedback on ocean temperatures, potentially reducing the relevant climate sensitivities (e.g., Dong et al., 2022). It is however unknown if, and how strongly, one could tie global warming to ice sheet melt as a practical matter. This is something that could be explored in an ESM-ISM under idealized conditions and perhaps a relationship (and its uncertainties) derived between Antarctic and Greenland mass loss and global mean surface temperatures. Conceivably, one could use the ISMIP models to build such a parameterization (e.g., based on results such as shown in Fig. 1 of Edwards et al. (2021)) and, by exploring the uncertainty in that relationship, assess the uncertainty in ECS/TCR/TCRE due to inclusion of these processes. One wrinkle might be that the historical rates of freshwater input might not be coherent with

the parameterized scenarios due to intersecting effects of other forcings (e.g., aerosols, ozone depletion), internal variability or inadequacies in the models themselves.

Another alternative might be to use the meltwater directly from idealized runs with an ESM-ISM, however, there are significant conceptual difficulties in producing a stable and realistic ice sheet component for the pre-industrial era, and that may preclude this approach for the time being.

## 615 **5.4 DAMIP simulations**

Questions related to the detection and attribution of climate change have generally been covered by the Detection and Attribution MIP (DAMIP) protocols (Gillett et al., 2016) (and/or the Large Ensemble Single Forcing MIP (LESFMIP) (Smith et al., 2022)), which call for single or grouped subsets of forcings to elucidate the impact of, for instance, greenhouse gases, aerosols, natural and/or anthropogenic forcings. These are all counterfactual experiments that do not correspond to the real world, and as such, it would not be consistent to use observed anomalous FW flows unless we were certain that those trends were themselves cleanly attributable. An important use of this class of experiment is to decompose the results in the all-forcing simulations into a (possibly interacting) sum of the parts. Thus the freshwater forcing would have to appear somewhere in the protocol, perhaps as an independent phenomenon. For instance, if one assumed that the trends in anomalous FW were purely anthropogenic (i.e., they would not have occurred without human interference in the climate system), they would be used in the anthropogenic-only simulations, but not in the natural forcing only runs. However, without a huge amount of ESM-ISM experimentation (or the use of a parameterization as described above), the individual impacts on the flux from greenhouse gases, aerosols, ozone trends, or natural forcings are as yet unquantified.

Note that freshwater is not the only forcing for which this ambiguity exists. The role of changes in biomass burning in historical simulations (which is based on observations) has a similar issue, and was effectively assumed to be purely anthropogenic in DAMIP in CMIP6 since no changes were included in the *hist-nat* simulations (Gillett et al., 2016). Similarly, dust emissions are not generally treated as a forcing at all, despite the clear mismatch between models and observations in the variation in dust emissions over time (Kok et al., 2023). A better approach to these examples might be to assess the natural changes with an interactive module (for fire or dust or ice sheets), with estimate of the anthropogenic components derived as a residual.

## 5.5 Future Scenarios

In the absence of future observations, future scenarios will require modeled freshwater inputs (Knutson and Tuleya, 2005). These scenarios could be taken from existing ISMIP6 simulations (Nowicki et al., 2020; Payne et al., 2021), new ISMIP7 simulations, or from existing or upcoming coupled ESM-ISMs output (e.g., Schloesser et al., 2019; Smith et al., 2021; Siahaan et al., 2022), though this may require an iterative process. For Antarctica, some historically-calibrated estimates of freshwater forcing and its associated uncertainties have been derived from ice-sheet model projections spanning 1990 to 2300 under two SSP scenarios (Coulon et al., 2024). These projections, provided at annual resolution for 27 drainage basins, suggest that the total freshwater flow from the Antarctic ice sheet could increase up to fourfold by 2300 under an extreme climate scenario. They also indicate that the partitioning between icebergs, basal melt and runoff, which aligns well with observational estimates

(Davison et al., 2023) over the historical period, is expected to change substantially in the coming decades and centuries, especially under extreme warming. In both climate scenarios examined, sub-shelf melting increases, altering the form and location of freshwater flow. However, there is considerable uncertainty in future ice sheet changes, and a key consideration in designing an appropriate range of scenarios for the freshwater inputs will be the need to encompass the structural uncertainty in the ISMs themselves, as well as the scenario and climate sensitivity dependencies in any specific dataset. This should be a high priority for the community to assess, but is beyond the scope of this paper. At minimum, scenarios continued from the historical simulations should continue with constant fluxes, as opposed to abruptly setting them to zero.

Beyond the standard storyline scenarios (the SSPs or RCPs), new ideas for a "What-If MIP" have been proposed (WCRP, 2025) which would focus on the climatic consequences of large tipping point events, such as a collapse of the WAIS. Since this kind of event would have large consequences for the freshwater budget in the Southern Ocean, some thought should be given to defining a plausible freshwater forcing scenario to go along with the reduction in ice sheets. Again, developing this is beyond the scope of this paper. One such "What If" future scenario is based on a 95th percentile projection for the GrIS from a structured expert judgment exercise (Bamber et al., 2022).

#### 5.6 Paleoclimate simulations

The Paleoclimate Model Intercomparison Project (PMIP) has focuses on the fidelity of climate model simulations for key paleo-climatic periods – notably the mid-Holocene (6 ka: *midHolocene*), Last Glacial Maximum (21 ka: *lgm*), last interglacial (127 ka: *lig127k*), the mid-Pliocene Warm Period (3.2 Ma: *midPlioceneEoi400*), and also the last millennium (850 CE to present; *past1k*), and provides 'out-of-sample' tests to judge the credibility of historical and future simulations (Kageyama et al., 2018; Zhu et al., 2021; Schmidt, 2010; Schmidt et al., 2014). Most of these experimental designs are equilibrium (time-slice) experiments, and as with the pre-industrial controls described above, the models can be configured such that the freshwater balances the net accumulation over the ice sheets. However, the location of the ice sheets, the discharge, and the partitioning of the discharge between icebergs, basal melt and runoff may be quite different than for the *piControl*.

For *lig127k, midHolocene* or the *past1k*, the *piControl* spatial distribution of FW is probably adequate, since the ice sheet geometries specified in their respective protocols are very similar to the present-day (Kageyama et al., 2018). However, the *lgm* or any deglaciation experiments will include Laurentide and Eurasian ice sheets, as well as expanded Greenland and Antarctic ice sheets, while the *midPlioceneEoi400* protocol uses ice sheets that are smaller than present. Estimates of freshwater forcing for these ice sheet states could be derived from ISMs run for their respective periods, and the spatial distribution of the iceberg melting estimated either from ice-rafted debris maps or, eventually, iceberg-enabled high resolution paleo-ocean simulations (as used above). Increasing interest in last interglacial ESM simulations with retreated WAIS (e.g., Hutchinson et al., 2024; Berdahl et al., 2024) will also require new FW estimates.

There are a number of considerations specific to such long time periods that should be noted:

- Ice sheet models typically output spatial fields as snapshots at relatively low temporal resolution, and scalar diagnostics (e.g., quantities integrated across a whole ice sheet) at higher temporal resolution. Since calving mass flow rates are very

temporally variable, a single snapshot is not suitable for prescribing calving mass flow rates in subsequent climate modeling. Instead, a long-term average is needed, which will most likely be derived from a scalar time series. Unfortunately, this means that the spatial distribution of calving is unlikely to be available unless requested ahead of time.

- Time-slice ESM simulations require balanced water mass flow rates, i.e., ice sheet freshwater must be balanced by accumulation. Ice sheet response to climatic changes can take several thousands or tens of thousands of years (e.g., Garbe et al., 2020; Noble et al., 2020). Consequently, ice sheets are unlikely to be in steady-state during periods of interest such as the last interglacial when climatic changes were relatively fast. The ESMs may therefore need to adjust the total freshwater inputs as a function of their own biases in snowfall and ice sheet mass trends.
- There will be a choice of several ice sheet simulations for particular regions or periods, and likely some inconsistency in data availability or reporting. Selection or weighting of individual ice sheet simulations could be based on their consistency with geological reconstructions during the period of interest ideally in a framework that accounts for considerable uncertainties in both simulations and reconstructions (e.g., (Kageyama et al., 2021)).
  - Paleo ice sheet simulations generally run over long time periods and start from a spin-up, in contrast to modern simulations starting from data assimilation or a nudged spin-up. This means a present-day ice sheet state reached at the end of a paleo simulation is generally not as good a fit to present-day observations (geometry and magnitude), than would be achieved by data assimilation. Users may need to decide whether to use forcings as anomalies from the simulated present-day state, or as (potentially more biased) raw values.
  - Fresh water mass flow rates derived from transient ice sheet simulations will reflect ice sheet geometries that will not necessarily match the ice geometry for the respective paleoclimate modeling protocol. Representative ice sheet geometry data will need to be provided with the freshwater forcings, so that users can assess for themselves whether there are important differences.

Transient simulations, such as the deglaciation, the 8.2kyr event, or Heinrich events (with the massive inferred expansion of the iceberg meltwater), require more specific efforts (e.g., Fendrock et al., 2023). Reasonable estimates of changing freshwater flows could be made for these periods, but more informed assessments will require a more concerted effort bringing together paleoceanographers and modelers.

#### 6 Conclusions

Over the longer term, it is clear that the community needs to move faster towards coupled ESM-ISM models and we are optimistic that progress is being made. However, in the absence of this capability across the multi-model ensemble, and the need to track the structural uncertainty in these simulations, treating freshwater inputs and changes as a forcing will likely be useful. We acknowledge that the spatial distributions, breakdown of discharge, and depth profiles recommended above are gross simplifications, and individual fjords, ice shelves, or calving events may inject freshwater in vastly different locations,

depths, or time. Nonetheless, given the need to represent ice sheet freshwater export in CMIP-class models whilst ESM-ISMs are under development, this data provides a first approximation for including these factors in existing models.

We note that the definition of the time-series and the implementation of the FW forcing into ocean models are separable, and
the suggested interface to the ocean for these fluxes would be applicable even to interactively calculated mass flow rates in the
absence of a prognostic iceberg parameterization. Similarly, this framework will allow for reruns and testing of different ESMISM generated historical or future mass flow rates in an analogous way to the use of AMIP simulations or fixed-composition
simulations instead of fully-coupled oceans or fully-interactive composition simulations, which are much more expensive or
complicated to run.

We provide regionally disaggregated time-series of freshwater forcing estimates for all major basins in Greenland for 1850 through 2024 and Antarctica for 1990 through 2024, along with estimates of the uncertainty, and 3-dimensional spatial profiles of meltwater input into the ocean via both ice shelf submarine melt and iceberg melt. The products are designed to be flexible and adaptive to specific choices that individual modeling groups make—for instance, the two types of approach to climatological ice sheet freshwater forcing, or decisions to partially resolve some ice-shelf cavities but not others. We also provide hemispherically and ice-sheet averaged equivalent fields for simpler implementations. Moving forward, these data will be updated annually, hopefully within 3 months of the end of the calendar year.

Evaluation of the impact of this forcing on model simulations will take some time, though increasing the spatio-temporal network of ocean observations i.e., in situ and remote observations, will improve estimates of the impact of current meltwater input. This requires a concerted effort to better integrate the suite of ocean observing networks and ensure their longevity so that changes can be quantified robustly.

725

Code and data availability. Data is available at https://doi.org/10.5281/zenodo.14020895 (Mankoff et al., 2025a) and code is available at https://doi.org/10.5281/zenodo.15707384

Table A1. Paraphrased questions and summarized model group responses

| Question                                   | Responses                                    | Notes                                         |
|--------------------------------------------|----------------------------------------------|-----------------------------------------------|
| Mass or volume-conserving ocean model?     | Mass: 8; Volume: 8                           | One will switch from volume to mass in CMIP7  |
| Natural boundary conditions or             | Natural: 9; Equiv. FW: 7                     |                                               |
| equivalent freshwater fluxes?              |                                              |                                               |
| Closed piControl water mass budget?        | Yes: 14; No: 2                               |                                               |
| Closed piControl energy budget?            | Yes: 4; No: 8; Partially: 4                  | Of the yes/partial responses, latent heat was |
|                                            |                                              | conserved, but few models kept track of the   |
|                                            |                                              | sensible heat. No models considered potential |
|                                            |                                              | energy of the snow.                           |
| Discharge in historical simulations?       | Fixed at <i>piControl</i> level (Type 1): 5; |                                               |
|                                            | Mass balance assumed (Type 2): 6;            |                                               |
|                                            | Ignored: 2                                   |                                               |
| Spatial pattern for iceberg melt?          | Global/Latitudinal bands: 3;                 |                                               |
|                                            | Ice-shelf adjacent: 7;                       |                                               |
|                                            | Pre-calculated map: 2;                       |                                               |
|                                            | Lagrangian icebergs: 2;                      |                                               |
|                                            | Nothing: 2                                   |                                               |
| Sub-ice shelf cavities in the ocean model? | Yes: 0; No: 16                               |                                               |
| Historical increases in mountain           |                                              |                                               |
| glacier melt as a potential forcing?       | Yes: 2; No: 10; Maybe: 4                     |                                               |

Modeling groups/models that responded: ACCESS CSIRO, CCCma/CanESM5, CESM, CNRM-Cerfacs, Fondazione CMCC, GISS ModelE, HadGEM3-GC3.1, IITM-ESM, INM, IPSL-CM, MIROC, MRI, Nanjing University IST-ESM, NCC-NorESM, UKESM, U. of Arizona

## Appendix A

In January 2024, the organizers of the workshop sent a questionnaire to all CMIP modeling group contacts asking about their model's practice for dealing with cryosphere-related freshwater flows. We received 16 responses (out of approximately 30 groups). A condensed summary of the questions relevant to this paper and the responses we received is in Table A1. Not all questions were answered by all groups, and there are some subtleties in the responses that are not captured in this summary.

Author contributions. GAS led the push to consolidate this effort and led the writing. KM contributed to the concept and did the data processing and editing. DR created figures 1 and 6. JLB, CB, DC, DMC, VC, BJD, MHE, PRH, NCJ, QL, JMM, PM, CRM, TAM, RM, SN, AOA, AGP, TR, and DR all contributed important insight, data, and expertise, and edited the paper.

Competing interests. The authors declare no competing interests.

Acknowledgements. We'd like to thank the participants in the February 2024 virtual workshop on Anomalous Freshwater Forcings for their input (particularly Julie Arblaster and Xylar Asay-Davis) the CMIP panel for supporting this effort. GAS, KM, QL, and DR were supported by the NASA Modeling, Analysis and Prediction program. PM and NCJ received funding from Agence Nationale de la Recherche 740 - France 2030 as part of the PEPR TRACCS programme under grant number ANR-22-EXTR-0010. TR has been supported by the European Commission Horizon 2020 Framework Programme nextGEMS, H2020 Societal Challenges (grant no. 101003470), MHE acknowledges support from the Australian Research Council (ARC Grant Nos. SR200100008, DP190100494). RM, PRH, DMC, VC, NCJ and AOA are supported by OCEAN ICE, which is co-funded by the European Union, Horizon Europe Funding Programme for research and innovation under grant agreement Nr. 101060452 and by UK Research and Innovation (Internal O:I Contribution number 22). TAM was supported by 745 U.S. National Science Foundation Office of Polar Programs grant 2052551. DC was supported by U.S. National Science Foundation Office of Polar Programs grant 2052549. JLB was supported by European Union's Horizon 2020 research and innovation programme through the project Arctic PASSION (grant number: 101003472) and from the German Federal Ministry of Education and Research (BMBF) in the framework of the international future AI lab "AI4EO - Artificial Intelligence for Earth Observation: Reasoning, Uncertainties, Ethics and Beyond" (grant number: 01DD20001). BJD was supported by ESA through the Polar+ Ice Shelves (ESA-IPL- POE-EF-cb-LE-2019-834) 750 and SO-ICE projects (ESA AO/1-10461/20/I-NB), and through NERC awards NE/T012757/1 (DeCAdeS) and NE/Y006291/1 (NSFGEO-NERC: Investigating the Direct Influence of Meltwater on Antarctic Ice Sheet Dynamics [NSF award number 2053169]). We would also like to thank John Dunne, Cecilia Bitz and an anonymous reviewer for their constructive comments on an earlier version.

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
