# Peer review of "Datasets and protocols for including anomalous freshwater from melting ice sheets in climate simulations"

_EGUsphere, 2025_

## Author Comment (AC1)

**Response to Reviewer 1 (John Dunne):**

Reviewer comments are red font. Replies are in black font.

The manuscript "Datasets and protocols for including anomalous freshwater from melting ice sheets in climate simulations" by Schmidt et al provides an interdisciplinary assessment of the state of understanding, uncertainties, and many technical issues involved in representing the transient liquid and solid freshwater forcing from the Greenland and Antarctic ice sheets in coupled climate Earth system models as efforts to couple these models with ice sheet models continue to advance.  It certainly makes the case that this is a very technically complex problem and progress on the interim solution of applying freshwater anomalies is an important step in improving the representativeness of the ocean circulation and climate responses to anthropogenic climate change in general and the recent observed changes to ice sheet mass balance.

Thank you very much for your support and constructive review.

My main issues with the present version are:

- The fluxes for Antarctica equivalent to Figure 3 for Greenland are not presented

  We have added figures for all data products.

- The overall value of representing these fluxes to improve representation of sea level rise is not provided… how much of non-steric sea level rise will this provide?

  We've added a paragraph on the impacts of these fluxes on the representation of sea level in Section 3.

- There are many occasions of handwavy statements of the status quo being insufficient without clear recommendations to relieve the problem. I have pointed these instances out and made suggestions.

  Thanks. We have tried to be clearer in our recommendations.

See technical suggestions below.

Line 2 – add comma before "and"

Done

Line 5 – add "over the historical period" after "discharge", replace "accounted for" with "addressed" or "incorporated" or "represented", and remove "an updateable dataset of"

Done. New phrasing is "but in neither case was the observed increasing discharge over the historical period properly represented. In this paper, we present data products of absolute and

anomalous freshwater mass fluxes from both ice sheets, and recommendations for their use in historical simulations."

Line 14 – doesn't "over the last century" include "and in recent decades"? Suggest removing or changing to "and has accelerated in recent decades"

Added "accelerated" suggestion.

Line 30 – Another example of the role of freshwater on Southern Ocean circulation

Bronselaer, B., Winton, M., Griffies, S.M., Hurlin, W.J., Rodgers, K.B., Sergienko, O.V., Stouffer, R.J. and Russell, J.L., 2018. Change in future climate due to Antarctic meltwater. Nature, 564(7734), pp.53-58.

Added.

line 63 - There does not seem to be consistency between the definition of "Discharge" here (as an ice flux) compared to the terms in Figure 1 (e.g. "Discharge" not identified but "Subglacial Discharge" seemingly identified as a liquid flux)… is "Discharge" equal to "Iceberg Flux" or to to the sum of several terms in Figure 1?

We have improved Fig 1 and made the definitions consistent.

Line 72 – "Ice front" is not provided in Figure 1, but "Frontal retreat" is provided twice.

Fixed.

Line 74 – "Ice shelf" is not provided in Figure 1

Added.

Line 114 – ", and again this is with respect" should be "relative"

Done, and nearby "with respect to" are also changed to "relative to".

Line 124 – "uasi" should be "quasi"

Done.

Line 129 – I don't think the sentence, "The implications of adding a new forcing dataset needs to be considered for each of these different configurations and experiments." Is helpful without further contextualization and should be removed unless these implications are to be detailed.

We have explicitly linked this statement to the relevant section where we discuss this.

Line 160 – The GFDL CMIP6 models included explicit icebergs: Adcroft, A., Anderson, W., Balaji, V., Blanton, C., Bushuk, M., Dufour, C.O., Dunne, J.P., Griffies, S.M., Hallberg, R.,

Harrison, M.J. and Held, I.M., 2019. The GFDL global ocean and sea ice model OM4. 0: Model description and simulation features. Journal of Advances in Modeling Earth Systems, 11(10), pp.3167-3211.

Added: "to explicit modeling of icebergs (Adcroft et al., 2019)"

Line 181 – add comma before "it"

Disagree, no change.

Line 185 – "observational changes" should be "observations" to avoid repeating "changes"

Changed.

Line 295 – Before moving on, it would be helpful to know how these freshwater fluxes compare to those in Swart et al., 2023 for SOFIA which is cited earlier.

The SOFIA protocols are substantially more idealized than the fluxes given here, which makes sense. They are interested in doing some additional experiments with these historical forcings, but this is still to be determined. We've put in some text alluding to that.

Line 310-312 – Is "with respect to the 1850–1900 pre-industrial period" mean that 1850-1900 is the "baseline period"?  I think so from line 102, but it is not clear why different wording is being used for these two things if they are indeed the same thing.

We have now reserved the term "pre-industrial" for the standard PIcontrol setup. The baseline period is now clearly 1850-1900 (for Greenland) and is not described as pre-industrial.

326 – remove "the choices available in"

Done.

Line 402 – "1.5 orders of magnitude" should be "50 times"

Done.

Line 404 – remove ", such as that provided by"

Done.

Line 412 – a good reference for "estuarine box model" is

Sun, Q., Whitney, M.M., Bryan, F.O. and Tseng, Y.H., 2017. A box model for representing estuarine physical processes in Earth system models. Ocean Modelling, 112, pp.139-153.

And

Sun, Q., Whitney, M.M., Bryan, F.O. and Tseng, Y.H., 2019. Assessing the skill of the improved treatment of riverine freshwater in the Community Earth System Model (CESM) relative to a new salinity climatology. Journal of Advances in Modeling Earth Systems, 11(5), pp.1189-1206.

Added.

Line 434 – add comma before "and" Not done.
Line 449 – add comma before "and" Done.
Line 450 – remove comma before "and" Done.
Line 451 – add comma before "or" Done.
Line 515 – add comma before "but" Done.

Line 515-517 – I do not understand the statement "One could remove an equivalent mass of deep water at the continental boundary to match the mass of freshwater coming from the floating source to allow the freshwater fluxes to be accurate, while also matching the sea level rise." Is the assumption here that models have rigid lids and virtual salt fluxes? This suggestion would not seem to appropriate with models that use a free surface and real freshwater fluxes.

We don't think this is correct. The issue is whether the model will correctly adjust for a change in the floating mass of ice shelves. Even with a free surface and natural freshwater fluxes, if there is no ocean under the ice shelf, it would need to be corrected for. Also if the pressure of the ice shelf on the ocean was fixed (or the cavity treated as a rigid space), this term would also be missing. For a model with a rigid lid (and constant volume), the sea level changes would not be represented properly in any case. We have clarified in the text under what circumstances this might be needed in the added discussion of sea level implications in Section 3.

Line 537 - add "spread" before "uniformly"

Done.

Line 557 – The statement "Increasing meltwater can act as a negative feedback on ocean temperatures, potentially reducing the relevant climate sensitivities" has two parts that have opposing influences on ocean stratification and surface warming response – the extraction of ocean heat and buoyancy for warming and melting of ice, and the addition of buoyancy from freshwater. I don't think it helps to combine them as a single statement unless one effect strongly outweighs the other on density depending on the fraction added as liquid.

We think that in practice, modeling has shown that the net effect is likely to be a negative feedback (see Schmidt et al (2023) and other cited papers). But we have rewritten this sentence to leave that open.

Lines 576-581 – These sentences should be restructured as explicit guidance rather than as a set of hypotheticals, i.e. "For models that do not include explicit ice sheets, we propose that freshwater forcing be included as part of the anthropogenic suite of forcings"

We don't think that this should be so definitively recommended. It is an open science question to be explored. We are not in a position in this paper to give a ruling on how much of the anomalous flux is anthropogenic, probably a lot, but it is hard to be certain.

Line 585 – Indeed, the GFDL models both had interactive dust in CMIP6.

This isn't quite what is meant. Interactive dust generally allows for varying dust emissions as a function of soil wetness, wind, etc. But there are additional anthropogenic sources of dust (from agricultural, recreational and construction activities) which can be inferred from the observations, but have not yet been incorporated into historical simulations. Conversely, the actual record of biomass burning is incorporated, but there is no split between natural and anthropogenic components.

Line 589-608 – I am surprised that no recommendation is made here to at least maintain the same freshwater fluxes from the end of the historical run through the future projections. This would seem preferable to using no freshwater fluxes. It would seem ill-advised for models to try to participate with these historical freshwater fluxes without having a plan for what to do with those fluxes through the transition to projections.

Agreed. We have added this to the section.

Line 599 – add comma before "and"

Done.

Line 613 – I would rephrase "judge the credibility of future simulations" as "judge the credibility of historical and future simulations"

Done.

Line 619 – remove "the" before "very"

Done.

Line 631 – Need to add something like "we therefore highly encourage ice sheet models to same long term averages"

Assuming that this was meant to read "save", we have added that to the sentence.

Line 635 – Need to add the implication of this long term imbalance for the provision of forcing, required length of ESM simulation, or otherwise… "We therefore highly encourage…"

Added.

Line 639 – Helpful here would be a recommendation for observational reconstruction references that should be considered as helpful to provide these constraints.

This is a moving target since these reconstructions are also being updated regularly. We have added a reference though.

Line 644 – is there a recommendation to be made here?

Not really, it's just a word of caution.

Line 654-657 – These two sentences are wandering and hand-waving.  How about, "While the community makes long term progress on explicit coupled ESM-ISMs, there remains urgent need to make near term progress with interim configurations treating freshwater anomalies as external forcings."

Replaced.

---

## Author Comment (AC2)

**Response to Reviewer 2 (Cecilia Bitz):**

Reviewer comments are in red font. Replies are in black font.

This paper provides a valuable service by outlining the range of ways that ESMs model the effects of ice sheets and glaciers and by constructing datasets and recommendations to deal with their mass and energy imbalance. There are many positive aspects, like the community effort and presumably consensus behind the work.

Thank you very much for your support and constructive review.

The list of definitions at the start is very good too - though I suggest adding Surface Mass Balance (SMB) to it.

Done

I am grateful to know where these authors "judge" something to be true or offer views. Some of this information would be impossible to know for certain since not all modeling centers answered their survey. The authors provide valuable guidance about simulation assumptions and practices with a summary of choices made by modeling centers with some indication of the consequences. This is helpful for model developers and those analyzing models alike.

Agreed!

Please consider the following specific comments:

1) There are a few occasions when the manuscript needlessly goes into minor issues about conservation. For example, I think the point made about moving snow down hill at line 166 is ridiculous, or needs further explanation. We don't worry about the conversion of PE to heat by raindrops when they hit the ground so why should we worry about the conversion of PE to heat when snow moves downhill. Further if it amounts to 14C it is not a wrinkle that might make a small difference, so something is awry about this paragraph. Another example is at Line 461-3 with regard to icebergs possibly melting a bit due to geothermal heating, etc. While all this may be true, there is little point in itemizing a bunch of stuff that no ESM developer would bother with. It makes the paper longer than it need be.

Fair points, but we are trying to cover all the bases/interests of a diverse group of authors (and readers!). Conservation is important in coupled modeling, and conservation of energy can be pretty subtle in atmospheric models - for instance, many models don't properly account for the specific heat of condensate, nor the changes of potential energy and so rain always falls at 0ºC even in the tropics! As models improve, these assumptions will likely be modified, and so while most coupled models don't currently include PE, that won't be the case forever. (PS. The 14ºC number comes from the simple conversion of $gh = c_i \, \mathrm{delta}(T)$, with h=3000m, g=9.81 m/s2, $c_i$=2090 J/kg so $\mathrm{delta}(T)$ = 9.81*3000/2090  = 14.1ºC though the total energy flux in W/m2 is

only about 0.2 W/m2 additional forcing over the ice sheets). These details are included so that we can flag these issues as potentially coming into play at some point.

2) It is good to see some discussion of how transient additions of freshwater will alter the ocean salinity and sea level. Yet, I thought this could be a teachable moment to explain the pros and cons of adding freshwater in volume vs mass conserving ocean models (rather than an oblique reference to there being a difference at line 162). I was intrigued by the discussion that volume conserving models would overestimate sea level changes since most recent ice loss has been from floating ice and models don't do hydrostatic balance of ice correctly. It seems like this should be elevated to a recommendation at the end for modelers to work on.

We agree. However, while we have our preferences we don't think it's the place of this paper to tell modeling groups what ocean models to use. We have added a new subsection in Section 3 to deal with sea level representation.

3) In Fig 1 the Antarctic side shouldn't have surface runoff and much ablation zone at the surface. It is strange to see so much of Greenland below sea level and so little of Antarctica. There are elements that seems reversed between Greenland and Antarctica. Is this really meant to be the final version of this figure?

We've updated the figure to account for this.

4) The figure captions could be improved with citations of sources and a bit more explanation of what the reader should glean from them. An example is Figs 4 & 5 where weighting functions are not defined either in the captions of main text. I would have thought a weighting function would be unitless. The caption should say where such maps are available.  A few pages further into reading the manuscript I see the term "Iceber met maps" in the main text but without reference to Figs 4 & 5. Given the units are the same I'm guessing this is what is plotted in Figs 4&5. If so, be sure to reference the figures, use the correct term in the caption, and state in the caption that this is one of your datasets.

Figure captions have been rewritten to provide more explanation of the figure, and explicitly state (when appropriate) that this data is supplied as part of this work with a link to the Zenodo data URL.

Another is Fig 3 which describes data with no citation. Is this one of the products that the authors are providing?

We have added new figures for every data product and explicitly point out these are products provided as part of this work.

If so, I'm concerned about the sudden switch in methods and variability at year 1986 since I expect some users will just prescribe this as forcing to their model without being aware of it and then "discover" an amazing regime change in their model in 1986, similar to the many discoveries in AMIP runs forced with sea ice concentration from HADISST. I recommend providing two separate products: One of the longer period with the same method throughout

and a shorter one since 1986. (Again a few pages further into reading the manuscript I see Fig 3 discussed in the main text, which is good. Please at least say in the caption that this is one of your datasets though.)

There are two things going on here, one was described in the submitted ms, the other was not well described. There is both a methods change and a system change near 1990. Specifically:

*The source data (Mankoff, 2020) has Greenland-wide spatial resolution from 1840 through 1985 and regional spatial resolution from 1986 onward (Mankoff, 2020a). To provide regional resolution for the entire time series we take the average of the earliest five years of regional resolution (1986--1990) to determine the relative contribution of each region to the whole, and then split the whole by that proportion from 1840 through 1985.*

The system change that occurred near 2000 is retreat (Greene, 2024). Zooming in on Fig. 3 (left) the system changes in 1986 - clearly due to methods. But that change appears to be only variability, not magnitude. The magnitude change (i.e., increase in discharge) occurs a few years later, near 2000, which is likely due to the system changing, not our methods.

We think the potential for dramatically different ocean responses as we transition to regional distinct fluxes from regionally uniform fluxes is small, but could be explored in future work.

5) Figure 6. I don't get why the circled 2 is called an "implicit FW" flux and the term "implicit SMB" is used at line 504. What is meant by "implicit" here and elsewhere (the term is used a lot)? It is supposed to mean implied, but I think maybe it is being used to mean derived or diagnosed from other quantities. It seems to me that no prognostic calculations in a model are implicit, so I can't grasp what implicit is supposed to mean.

"Implicit" in these situations means that this is calculated as a consequence of other factors rather than being a bottom-up parameterization or calculation. (2) is referred to as an implicit flux into the ice sheets because there is no explicit ice sheet reservoir that is being passed the local impact of the SMB. Fig 10. (now) is unchanged.

Also in Figure 6, I would think circled 2 should be labeled SMB rather than FW flux since FW flux is not as specific. The lower panel's cartoon of ice sheet to ocean fluxes is unclear and seems inconsistent with the caption. A cartoon should be an aid not a head scratcher.

The net impact of the SMB is a FW flux, and we have now made that explicit in the revised Fig. 10..

6) Lines 111-114 are too hard to follow. Break this up into more sentences and or include a table/equations.

Rewritten as: *The sub-shelf melt anomaly comes from one or the average of Davison (2023) and Paolog (2024) when they overlap, after setting the baseline to 1997.*

7) Line 138-9 Are you saying that some models just disappear the approx 3300 Gt/yr of accumulation on Antarctica? I thought models at least dropped the mass/volume of water into the ocean, as discussed a few paragraphs later. I've never heard of a credible ESM that didn't conserve freshwater at this level at least.

Indeed. This came as a bit of a shock to us too. But at least two models ignore this completely. The resulting salinity drift is small and if groups aren't doing very long runs it might not show up very obviously, but regardless, this will bias the ocean circulation and stratification around the ice sheets. We do not recommend this practice (obviously).

8) The manuscript jumps between modeling approaches and dataset details a bit in somewhat confusing ways. For example, line 379 is probably about a dataset provided, or possibly it is about a particular model. Either way the text should clarify.

We have worked to make the text flow better.

---

## Author Comment (AC3)

**Response to Reviewer 3:**

Reviewer comments are red font. Replies are in black font.

"Datasets and protocols for including anomalous freshwater from melting ice sheets in climate simulations" By Schmidt et al. provides valuable transient forcing of ice sheets freshwater with protocol recommendations for modellers implementing the dataset consistently. This work certainly makes an important contribution to both the Earth system modelling and ice sheet modelling communities

Thank you very much for your support for this initiative and the constructive comments.

Please consider my suggested revisions below:

1. Missing figures showing the timeseries of each freshwater product listed in Section 3.3. These products are the key output of the paper. However, the features of some products are not presented and discussed. Figure 3-5 do this for Greenland ice discharge and iceberg maps, although Figure 5 is not cited in the main text. It would be good to include figures and discussions for the remaining products.

We have added additional figures for each data product..

2. Missing evaluations and discussions of how well this transient freshwater forcing and suggested modelling approaches improves the representation of ocean circulation, salinity, sea level, etc., in model simulations. The modelling approaches in the two runs are described in Section 4 (Pre-industrial control runs and historical simulations); however, evaluations of the methods and products are not provided. I appreciate the authors' time and effort, it would be good to include at least some preliminary modelled results using freshwater forcing from this work, along with comparisons to other studies.

Thanks. We are working on the implementation of these forcings in our own simulations but this takes time and would unnecessarily delay the publication of the forcing data if we were to wait until they were completed and analysed. We have made a conscious decision to only present the data inputs in this paper, in line with many other forcing input description papers for CMIP7. We have also added a section dealing specifically with sea level.

See specific comments below:

L5 – The abstract does not mention projections. Recommend clarifying that the proposed protocol relates solely to historical simulations but that freshwater forcing is likely to become very much more important in future projection scenarios.

We have clarified the abstract to be explicit about projections and have expanded Section 5.5 slightly to enhance that discussion. However, we aren't going to opine on the importance of the FW additions in the projections without having the projected fluxes ready to go - it may be that it

is a big deal for Greenland but relatively muted in the Southern Ocean (though Purich and England, Payne et al, Golledge et al have already looked at this). This is for a later paper in collaboration, for instance, with the ISMIP project.

L9 – Can the authors confirm that freshwater forcing will be included in the forcings for the CMIP7 historical experiments? CMIP7 is now very close to releasing this forcing, and it is unclear whether freshwater is actually included.

The goal is to provide an option for the CMIP7 simulations, but we are not in a position to mandate their use. We hope that a significant fraction of the historical simulations will employ these data.

L21 – Not convinced that computational expenses hamper the incorporation of ice sheets into ESM. They are by comparison to other elements of the climate system cheap.

This is true in the run-time sense, but the long time scales for ice sheets to equilibrate are prohibitive for CMIP-class models and so this has slowed the ability of such models to include ice sheets that are both accurate with respect to the present and with similar sensitivities to observed ice sheets, which could require multiple Holocene-length simulations.

Providing historical freshwater forcing is a step forward but it does introduce an issue around projections. What are ESMs supposed to do at the start of the projection period?

This is discussed in section 5.5.

L42 – There is no Appendix B.

Fixed.

L60 – Not convinced that these definitions are helpful. Are there not standardized definitions (CMIP or ISMIP6/7) that could be used? Runs the risk of making what is a fairly confused set of terms worse. Definitions could be improved by adding relevant units.

These definitions are for clarity within this paper, and are useful for that role. We have added discussion of the units and a definition for SMB.

L65 – This is confusing and does not match the definition of discharge immediately above (in which discharge sits between ice sheets and ice shelves, while here discharge is after ice shelves before calving).

Minor tweak to language for clarity.

L67 – Upstream of grounding line? What about ice floating on subglacial lakes? This is upstream but not grounded.

We have rewritten to be clearer that this means upstream of the final grounding line. Sub-glacial lake hydrology is significantly beyond the scope of this work!

L81 – Runoff is not purely a mass loss process if it includes rainfall (and melt of seasonal snow).

SMB is P-E-R, where R is the net runoff flux after any refreezing or retention, so we think that this can be considered a mass loss, just as precip is a mass gain. For a diagram of all mass flow terms, see https://github.com/mankoff/sankey (which is discussed in Mankoff (2025)).

This section may be helped by linking the definition to Figure 1 more tightly. At present, terms in Figure 1 do not match the definitions (eg basal melt) and vice versa (eg discharge).

We have improved the descriptions in the figure and expanded the definitions in the text to better align with one another.

L97 onwards – Defining symbols for these various quantities and how they relate to one another would be helpful here. There are a number of things going on in the section (baseline/anomaly, time period of baseline, separation of freshwater into various fluxes such as grounded, shelf melt and shelf calving) that need to be explained more clearly. There is also some danger in referring to 'ice mass anomalies' because terms such as this are often used in ice mass budget calculations (freshwater fluxes are related primarily to the loss terms where mass budget relates to the net of gain and loss).

We are now explicit in how we define 'anomaly' and are clearer in the definition of the baseline for any product (easy for Greenland, harder for Antarctica).

L100 – Needs to be made clear that GRACE does not directly measure freshwater flux.

This was not the intended implication, but the sentence has been adjusted to be clearer.

L104 – Figure 3 is cited before Figure 2.

Fixed.

L105 – This seems to be very detailed and relates to the way in which the forcing has been calculated. Would be better in Section 3 or 4 rather than Introduction.

We feel that the discussion of what we mean with 'anomalies' is fundamental to this paper and so we respectfully will leave this discussion where it is. We have nonetheless tried to make the discussion a little clearer.

L118 – Good to state that basal melt of grounded ice not included in the assessment however requires some justification to demonstrate, for instance, that this flux is much smaller than the other freshwater fluxes leaving the ice mass.

We now reference the summary paper of Manoff (2025) for that assessment (see https://github.com/mankoff/sankey )

L124 – quasi.

Fixed.

L153 – This is too vague – this assertion needs to be supported by references.

This assertion was summarised from the responses to our survey, and our own practices. We have put in some relevant citations.

L159 – Use of the term 'Discharge' here and in the next paragraph is confusing. Does this refer to the previous definition of 'discharge' or discharge of freshwater? If the latter, then it is not clear that the ESM calculations discussed here would explicitly calculate discharge (line 171).

This term refers to freshwater, and we have adjusted this paragraph. In the next paragraph, it is the discharge per the definition. They do not explicitly calculate discharge, but it is generally parameterized in order to account for non-dynamic ice sheets in GCMs.

L200 – Define what is meant by 'regional flow' here. Will this be done on the basis of sectors/basins as per IMBIE etc?

Edited to "we aim to provide data that represents the regional ice discharge, runoff, calving, and melting and their change over the historical period for…" And yes, per Fig. 2 we provide this at regional resolution based on the IMBIE etc. regions.

L203 – based on Rignot et al. (2013) and Mouginot and Rignot (2019).

Fixed.

L210 – Earlier grounded basal melt was explicitly disregarded.

We remove reference to basal sourced water here. There is water released throughout the year, but the bulk is released during the summer months.

L213 – Clarify differences between refreezing and retention. Retention on what timescales?

Removed retention - it was an unnecessary detail. Runoff is melt + rain - refreezing.

L223 – Why choose flux gates 5 km upstream from the grounding line?

We've added an explanation - but specifically it is because estimating ice thickness at the terminus is challenging, so flux gates are usually placed inland.

L242 – Clarification required here – grounding line retreat in itself is not a source of freshwater (increased freshwater flux and GL retreat are both consequences of increased melt for instance). Repeated confusion around inclusion of grounded ice melt.

Changed to "Additional sources of freshwater include frontal melt expressed as grounding line retreat…" and that basal melt is not included, "We neglect the basal melt of grounded ice because it is both one of the smallest terms (Mankoff, 2025) and it is mostly a steady state

processes, and therefore has no anomaly, which is the focus of many of the products presented here.

L245 – 'Surface runoff' Ideally all terms of this type should be defined in Section 1.1.

Removed most use of the word 'surface' as it was unnecessary.

L254 – Figures 3 and 4 are not cited in this section. The section does not include any discussion about the melt of ice bergs (i.e., Figure 4).

All figures are now linked in the appropriate sections, and all data products now have a specific figure. The iceberg melt distribution maps are explicitly discussed in Section 3, and in more detail in Section 4.1.2.

The discussion on p5 about baselines and anomalies is not mentioned at all.  It is unclear at this stage what form the freshwater forcing will be take.

The explicit statements about the form of the data products are in Section 3.3.

L262 – This is confusing Antarctic SMB has little to do with freshwater release from Antarctica and it is not clear what information on freshwater can be gleaned from trends in SMB.

The trends may not be relevant, but SMB is a key input to the mass balance of ice shelves. The method uses surface elevation and SMB inputs to estimate basal melting. We also use the input/output (IO) method to estimate grounded mass loss (most of which feeds into ice shelves), and the IO method uses MB = SMB - Discharge.

L266+271 – There are a few steps between the observations mentioned in the preceding paragraph and 'good estimates of Antarctic freshwater fluxes' that may be worth explaining.

Added: *We use the input/output method (mass balance = SMB - discharge) to estimate grounded mass loss (Rignot et al., 2019), a similar method to estimate ice shelf basal melt (SMB inputs minus elevation change, Davison et al. (2023); Paolo et al. (2024)), and remote sensing image time series of iceberg calving (Davison et al., 2023).*

L273 – Similar issue - sea level rise (ie net mass budget) is not directly linked to freshwater discharge (which is primarily to do with mass loss only).

We have added a specific section dealing with sea level in Section 3.

L289 – Is the ratio by each basin or the entire Antarctica?

Line removed for clarity since these calculations are independent. .

L297 – It would be helpful to have information on units here, as well as temporal resolution etc.

Added: *We provide the following data products. All time series are annual temporal resolution and units Gt yr$^{-1}$. All products have regional resolution per ice sheet (Fig.~\ref{fig:regions}). The iceberg melt maps 0.5 degree spatial resolution and monthly temporal resolution, but are steady state. Units for iceberg melt maps are m$^{-2}$, and when multiplied by cell area maps should sum to one.*

L297: Also it is unclear what the relationship is between the various products. For instance, were discharges and runoff used to calculate the freshwater anomalies for Greenland? Similarly, for Antarctica was calving and submarine melt used to find the freshwater anomaly? How? A conceptual figure may help here (also note use of products in protocol mentioned below).

We have expanded on the methodology in the text.

How is this related to the iceberg melt maps?

The iceberg melt maps are orthogonal to the calculations of the FW anomalies themselves.

These products are the key output of the paper. It would be useful to have some plots showing key features. Figures 4 and 5 do this for iceberg melt although they are not cited in this section.

Additional figures for each product have been added.

It would also be helpful to add some explanation of how each of the products is used in the protocols described in the rest of the paper and whether they are used as anomalies or absolute values. The two runs described in Section 4 are pre-industrial and historical: can the authors indicate which product is used when?

The piControl runs do not use any of our FW data - though they could use our maps, and protocols (for regional distributions and splits between forms of discharge). How the FW products are used in the historical runs is a function of the approach (as outlined in Section 4.3).

L351 – Tracer is not previously discussed.

Now mentioned earlier.

L353 – 'Relaxation time constant' seems to imply a specific way of incorporating freshwater into the pre-industrial but it is unclear what this method would actually be. More detail required.

The added flux (F) is defined as a weighted average of the discharge from the previous year (F0) and the previous year's ice sheet imbalance (I). The weighting determines the relaxation time constant. I.e. a weighting of 0.9/0.1 would be equivalent to a 10 year timescale. Written another way, F = F0 - (I - F0)/tau where tau is the number of years over which a change is effectively spread. The language in the text has been clarified.

L396 onwards – This appears to be a discussion around the fine-scale physics governing local redistribution of freshwater in fjord and around glacier fronts. Not clear how it relates to the implementation of the products in ESMs. Perhaps this text should be moved to an early section?

It relates because ESMs will increasingly resolve the larger fjords even if they didn't in CMIP6 - and the distribution of the discharge components will need to change accordingly. We are trying to be forward looking.

L421 – It would be good to have more discussion on the differences/consequences of the injection depth (surface vs depth) of freshwater from icebergs and ice shelves melting in the Southern Ocean.

Agreed. But we don't really know the answers yet, though the SOFIA project will make some progress related to whether it matters. We leave this for a future paper.

L429 – Not clear why the depth range of 130-230 m is suggested? '130 m is the mean depth of current ice shelf fronts in Antarctica.', but where does the number '230 m' come from?

We have expanded this section and now provide a rationale.

L434 – Please clarify if it is the configuration for the Greenland iceberg model? I don't think it is what was used in Mathiot and Jourdain (2023).

The Greenland iceberg modeling comes from Rackow et al (2017), updated by Marson et al (2024), which is a separate effort from that in Mathiot and Jourdain (2023) for the Southern Ocean, even though they are using the same base ocean model (NEMO).

L436 – The translation of iceberg fluxes to spatially variable melt rates is a key component of this paper and is lost here in the middle of a very detailed section. This important material would be better in its own subsection positioned to accompany Figures 4 and 5 perhaps between 3.2 and 3.3. Note that this text does not cite Figure 5. Currently, this text is located in the pre-industrial subsection although it would appear to be more generic than this and would be relevant to all time periods?

We have reorganised this into a section talking about general implementation and the specifics of a pre-industrial control.

L456 – Again, this subsection appears to contain a generic discussion around freshwater and energy. It is not clear why it appears under the pre-industrial.

As above.

L481 – Although this subsection is about the historical, the approach described (and shown in Figure 6) would appear to be applicable to projections as well (assuming suitable freshwater forcing was available).

Added.

L487 – May be mixing papers on historical and projection scenarios here.

Fixed.

L493 – It is more consistent to use 'Antarctica and Greenland' rather than 'SH and NH', as mountain glaciers are also included in SH and NH. Moreover, SH or NH acronym is not defined.

Fixed. But in the model results described the diagnostics were for each hemisphere, not the ice sheets specifically.

L508 – It is not clear how this discussion relates to the specific products that have been created. Which of the products are used and which are not? For instance, do the iceberg melt maps play a role?

This section has been reworked, with the sea level discussion moved to a separate section.

L589 – Worth pointing out that using this protocol for the historical implies that some extension of the freshwater forcing is required for projection experiments. Otherwise, there would be an abrupt change in forcing at the start of the projection period. An alternative might be to maintain the historical freshwater forcing into the projection period which would avoid a hiatus but become increasing unrealistic as (presumably) freshwater fluxes from the ice sheets increase into the future.

Yes. We have explicitly discussed this in Section 5.

L590 – Would it be possible to use this protocol with the variables supplied by ISMIP6? What additional work would be required? Presumably, as a minimum the iceberg melt patterns would need to be calculated for time in the future.

Yes. We are working (separately) to produce the relevant timeseries, but for the near term (say to 2100), we don't anticipate needing to change the iceberg melt patterns. But clearly, one could envisage extreme warming situations where the patterns will shift, but these will need to be calculated using appropriate models (that include Lagrangian icebergs, suitable melt parameterizations, and relevant projections).

L667 – This is a bit misleading. The products for Antarctica do not cover the period before 1991.

Fixed. New text: *We provide regionally disaggregated time-series of freshwater forcing estimates for all major basins in Greenland for 1850 through 2024 and Antarctica for 1990 through 2024.*

Figures

Figure 1. Indication of scale required for all panels.

We have improved the clarity and correctness of the figure, but note it's a schematic.

Depiction of runoff across and through the ice sheet on the Antarctica side of the diagram is misleading and should be on the Greenland side. If the diagram is meant to represent the contemporary ice sheets then the presence of a substantial ablation zone in Antarctica is also extremely misleading.

Agreed. Fixed in the updated figure.

Figure 4 – Please clarify how the iceberg melt rates are weighted in the caption and the main context. The same as Figure 5.

Fixed. New text: *Each map multiplied by cell area and summed equals one, so that these can be used for distributing freshwater inputs computed elsewhere.*

Figure 5 – What are the white lines in front of the Ross Ice Shelf in the subpanels [11] and [12]? Are they NaNs, or are they due to the grid mesh? If the latter, why are there no white lines in, for example, the panel [9], [10] and All?

Fixed.

---

## Author Response (AR2)

**Response to topical editor report**

Appendix name fixed.

All corrections to figures made as suggested.

SMB better defined.

Ocean module replaces ocean model where appropriate.

Thanks!